# Episodic release of $CO_2$ from the high-latitude North Atlantic Ocean during the last 135 kyr

Mohamed M. Ezat[1,2], Tine L. Rasmussen[1], Bärbel Hönisch[3], Jeroen Groeneveld[4] & Peter deMenocal[3]

Antarctic ice cores document glacial-interglacial and millennial-scale variability in atmospheric $pCO_2$ over the past 800 kyr. The ocean, as the largest active carbon reservoir on this timescale, is thought to have played a dominant role in these $pCO_2$ fluctuations, but it remains unclear how and where in the ocean $CO_2$ was stored during glaciations and released during (de)glacial millennial-scale climate events. The evolution of surface ocean $pCO_2$ in key locations can therefore provide important clues for understanding the ocean's role in Pleistocene carbon cycling. Here we present a 135-kyr record of shallow subsurface $pCO_2$ and nutrient levels from the Norwegian Sea, an area of intense $CO_2$ uptake from the atmosphere today. Our results suggest that the Norwegian Sea probably acted as a $CO_2$ source towards the end of Heinrich stadials HS1, HS4 and HS11, and may have contributed to the increase in atmospheric $pCO_2$ at these times.

[1] CAGE—Centre for Arctic Gas Hydrate, Environment and Climate, Department of Geosciences, UiT The Arctic University of Norway, 9037 Tromsø, Norway. [2] Department of Geology, Faculty of Science, Beni-Suef University, Beni-Suef 62511, Egypt. [3] Department of Earth and Environmental Sciences and Lamont-Doherty Earth Observatory of Columbia University, Palisades, New York 10964, USA. [4] Institute of Marine Environmental Sciences (MARUM), University of Bremen, Klagenfurterstrasse 2-4, D-28359 Bremen, Germany. Correspondence and requests for materials should be addressed to M.M.E. (email: mohamed.ezat@uit.no).

The ongoing rise in atmospheric $pCO_2$ and associated observations of reduced Arctic winter sea-ice coverage are projected to suppress the high-latitude North Atlantic ocean circulation over the coming decades, in turn affecting regional climate and the large-scale atmospheric circulation[1,2]. Regional reconstructions of past changes in surface ocean $pCO_2$ and temperature are important for understanding how climate, ocean circulation and the carbon cycle are linked. Greenland and Antarctic ice core records document a millennial-scale bipolar seesaw in air temperature changes during late Pleistocene glaciations and deglaciations[3]. Warm interstadial conditions over Greenland coincided with periods of gradual cooling over Antarctica, whereas cold stadial periods in Greenland coincided with warming over Antarctica[3]. In Greenland ice cores, these millennial-scale events have been termed Dansgaard–Oeschger events and are characterized by abrupt warming during the transitions to interstadials[4]. In contrast, Antarctic ice cores report only gradual climate changes[3]. The longest stadials include Heinrich events, and are called Heinrich Stadials (HS) (ref. 5). These interhemispheric climate patterns may be explained by variations in the Atlantic Meridional Overturning Circulation and associated changes in the northward heat export[6].

Atmospheric $pCO_2$ was $\sim$80–100 $\mu$atm lower during glacials compared with interglacial periods[7]. During the last deglaciation ($\sim$20–10 ka), atmospheric $pCO_2$ increased in two pronounced steps, by $\sim$50 $\mu$atm during HS1 ($\sim$18–14.5 ka) and by another $\sim$30 $\mu$atm during the Younger Dryas ($\sim$13–11.5 ka) (ref. 8). The last glacial period was furthermore characterized by millennial-scale variability in atmospheric $pCO_2$, with an increase of roughly 25 $\mu$atm beginning during most of the Heinrich stadials, and peaking at or less than a thousand years after the onset of the interstadials[9]. Thereafter, $pCO_2$ decreased gradually in phase with cooling in Antarctica[9].

The high-latitude North Atlantic, north of 50°N, is one of the most efficient $CO_2$ uptake areas in the modern ocean, because of cold sea surface temperatures, deep-water formation, strong primary productivity and high-wind speeds[10–12]. Therefore, it is an important region to study glacial-interglacial and millennial-scale variations in air-sea $CO_2$ exchange. This study aims to quantify the evolution of shallow subsurface ocean carbonate chemistry in the Norwegian Sea over the past 135 kyr, using the boron isotopic composition ($\delta^{11}$B) recorded in fossil shells of the polar planktic foraminifer Neogloboquadrina pachyderma. To constrain nutrient utilization, a primary control on the $pCO_2$ in the surface ocean, we also analysed Cd/Ca and $\delta^{13}$C in N. pachyderma. The study is based on sediment core JM11-FI-19PC retrieved from 1,179 m water depth in the Faroe-Shetland Channel (Fig. 1), in the main pathway of the exchange of surface and deep water masses between the Nordic Seas and eastern North Atlantic[13] (Fig. 2a). Our results suggest that the Norwegian Sea remained a $CO_2$ sink during most of the past 135 kyr, but during the latest parts of HS1, HS4 and HS11 the area acted as a source of $CO_2$ to the atmosphere. To elucidate the causes of these variations in seawater carbonate chemistry, we compare our results with previously published reconstructions of temperature[14,15], sea-ice cover, input of terrestrial organic matter and primary productivity[16].

## Results

### Geochemical proxies of ocean $pCO_2$ and nutrient changes.
Because the speciation and isotopic composition of dissolved boron in seawater depends on seawater pH, and borate ion is the dominant species incorporated into planktic foraminiferal shells, their recorded $\delta^{11}$B serves as a pH-proxy[17], and paleo-pH can be quantified if temperature and salinity can be constrained independently (see Methods for details). When pH is paired with a second parameter of the carbon system, aqueous $pCO_2$ can be estimated. Here we applied foraminiferal $\delta^{18}$O and Mg/Ca measurements to estimate foraminiferal calcification temperature and salinity, and then used the modern local relationship between salinity and total alkalinity to estimate coeval changes in total alkalinity (see Methods for details). Finally, we calculated the difference between our reconstructed shallow subsurface $pCO_2$ and atmospheric $pCO_2$ from ice core measurements[18].

The $\Delta pCO_{2sea-air}$ is a measure for the tendency of a water mass to absorb/release $CO_2$ from/to the atmosphere[10]. However, because N. pachyderma lives below the sea surface, this difference represents the difference between atmospheric $pCO_2$ ('air') and the seawater $pCO_2$ ('$pCO_{2cal}$') at the calcification depth and growth season of N. pachyderma ($\Delta pCO_{2cal-air}$). Neogloboquadrina pachyderma is thought to inhabit a wide and variable range of calcification depths in the Nordic Seas from 40 to 250 m water depth[19]. It migrates vertically in the water column[19] and is most abundant during late spring to early autumn[20]. To assess the influence of the seasonal occurrence and calcification depth of N. pachyderma on our results, we calculated $pCO_2$-depth profiles for the upper 250 m of the water column in the Norwegian Sea based on modern hydrographic data (total dissolved inorganic carbon, total alkalinity, temperature, salinity, phosphate and silicate) covering the late spring to early autumn[21] (Fig. 2b). The resulting modern $pCO_2$-profile (Fig. 2b) shows that the average $pCO_2$ of the surface ocean (0–25 m water depth) is 30–50 $\mu$atm lower than atmospheric $pCO_2$, but at the calcification depth of N. pachyderma ($\sim \geq$50 m water depth) average aqueous $pCO_2$ is approximately equal to atmospheric $pCO_2$. We thus calculated the difference in $pCO_2$ between the surface ocean and the atmosphere ($\Delta pCO_{2sea-air}$) by subtracting 40 $\mu$atm from $\Delta pCO_{2cal-air}$, assuming that the $pCO_2$ gradient between the surface ocean and calcification depth of N. pachyderma remained constant through time (see 'Discussion').

To characterize the changes in availability and utilization of nutrients, we measured Cd/Ca and $\delta^{13}$C in N. pachyderma. The Cd/Ca recorded by symbiont-barren planktic foraminifera such as N. pachyderma is sensitive to Cd concentrations in seawater[22], an element that shows strong similarity to the seawater distribution of the nutrient phosphate[23]. Thus, foraminiferal Cd/Ca can be used to reconstruct the levels of phosphate in seawater, and provides clues for the abundance and utilization of phosphate through time[24], albeit with a potential side control of temperature on the Cd incorporation into planktic foraminiferal shells[25]. In addition, planktic foraminiferal $\delta^{13}$C responds to changes in nutrient cycling, air-sea gas exchange, exchange between global carbon reservoirs[26] and carbonate chemistry[27].

### Seawater pH and $pCO_2$.
The studied sediment core JM-FI-19PC spans the last 135 kyr (refs 14–16) and has been correlated closely to the age model of the Greenland ice core NGRIP (ref. 28) (see Methods and Supplementary Fig. 1). The $\delta^{11}$B record displays $\sim$1.5‰ higher glacial values compared with interglacials and the core top samples. In addition, negative $\delta^{11}$B excursions of up to $-1.5$‰ occurred during HS1 and HS4 (Fig. 3a). Correspondingly, glacial pH was elevated by $\sim$0.16 units in the shallow subsurface compared with the Holocene, similar to results from earlier studies of tropical regions[29,30], but the record is punctuated by brief episodes of acidification during some Heinrich stadials (Fig. 3b). The reconstructed shallow subsurface $pCO_2$ shows lowest values of $\sim$200 $\mu$atm during the Last Glacial Maximum (LGM) ($\sim$24–19 ka), whereas it increased to 320 $\mu$atm during HS1 at $\sim$16.5 ka, and then gradually dropped to $\sim$230 $\mu$atm over the Bølling-Allerød interstadials (14.7–12.7 ka)

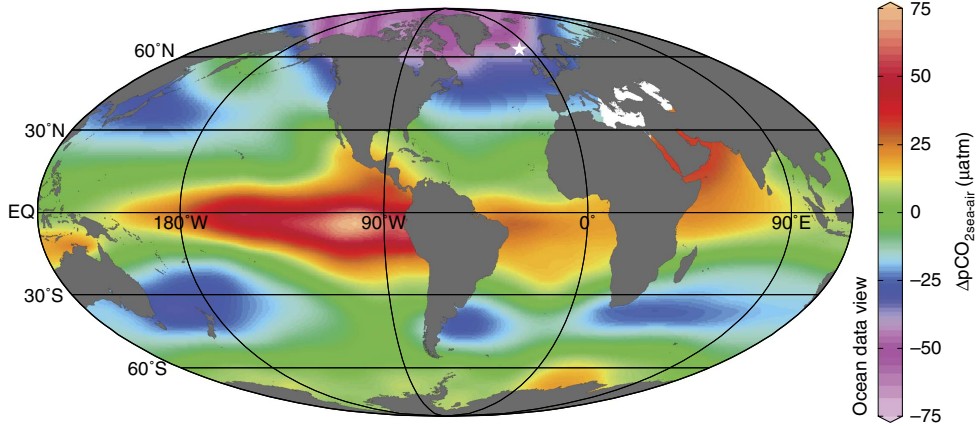

**Figure 1 | Map of mean annual $\Delta pCO_{2sea\text{-}air}$.** Oceanic $CO_2$ sinks and sources are presented by negative and positive $\Delta pCO_{2sea\text{-}air}$ values, respectively, and identify the high-latitude North Atlantic as a significant $CO_2$ sink. The white star shows the location of the studied sediment core JM-FI-19PC. Map was generated using Ocean Data View[66] based on modern data of Takahashi *et al.*[10]

(Fig. 3c). The $\Delta pCO_{2cal\text{-}air}$ increased from $\sim +5\,\mu atm$ during the LGM to $\sim +100\,\mu atm$ during HS1 (at $\sim 16.5\,ka$) and gradually decreased towards the Bølling–Allerød (BA) interstadial (Fig. 3e). Because of the analytical effort required for boron isotope measurements, and inadequate sample sizes for high-resolution boron analyses in some Heinrich stadials, we chose to focus on HS4 ($\sim 40$–$38\,ka$) as representative for the last glacial Heinrich stadials, because of the high sedimentation rate and good age control[14] on this interval in our record. The shallow subsurface $pCO_2$ increased from $\sim 220\,\mu atm$ during interstadial 9 ($\sim 40\,ka$) to $\sim 285\,\mu atm$ during HS4 and then gradually decreased to $\sim 225\,\mu atm$ during interstadial 8 ($\sim 37.5\,ka$) (Fig. 3c). Similar to the late part of HS1 ($\sim 16.5\,ka$) during Termination I, a prominent increase in the $\Delta pCO_{2cal\text{-}air}$ ($\sim +100\,\mu atm$) is also seen during the late part of HS11 (at $\sim 133\,ka$) in Termination II (Fig. 3e). A Holocene-like shallow subsurface $pCO_2$ is observed during the early and late Eemian interglacial (at $\sim 129$ and at $116\,ka$, respectively), but shallow subsurface $pCO_2$ was $\sim 30\,\mu atm$ lower during the mid Eemian (125–122 ka) (Fig. 3c).

**Cd/Ca and $\delta^{13}$C.** The $\delta^{13}$C record shows minimum values ($\sim -0.4‰$) during the Heinrich stadials HS1, HS3 and HS6, and $\sim -0.1‰$ during HS11, HS4 and some non-Heinrich stadials (Fig. 3h). The highest Cd/Ca values are recorded during HS1, HS11 ($\sim 0.007\,\mu mol\,mol^{-1}$), HS3, Younger Dryas ($\sim 0.004\,\mu mol\,mol^{-1}$) and HS4 ($\sim 0.0025\,\mu mol\,mol^{-1}$) (Fig. 3g). Although, the calcification temperature is found to have a secondary effect on the Cd incorporation into planktic foraminifera shells[25], the absence of a correlation between our raw Mg/Ca values, a temperature proxy, and Cd/Ca data ($R^2 = 0.0001$; Supplementary Fig. 2) supports the interpretation of the recorded Cd/Ca variability as changes in nutrient levels. However, it is notable that our Cd/Ca results show absolute values that are an order of magnitude lower than previous studies from the region[31,32]. We re-examined our Cd/Ca analyses closely and could not find any indication of analytical errors. The low Cd/Ca values can also not be attributed to the application of the intensive 'full cleaning' procedure to clean our foraminiferal samples before minor/trace element analyses (see Methods). Five duplicate samples of *N. pachyderma* cleaned with the standard cleaning protocol used in Cd/Ca studies yielded the same low Cd/Ca values (see Methods). Despite the low absolute values, our Cd/Ca data show strong consistency and agreement with the variations in $\delta^{13}$C values (Fig. 3g,h). In addition, our Cd/Ca trends are similar to previous studies, for example, similar Cd/Ca

for both the Holocene and the LGM are obtained as in Keigwin and Boyle[31] (Fig. 3g,h). As we cannot find the reason for the significantly lowered absolute values of our Cd/Ca, we refrain from quantifying the phosphate concentrations using Cd/Ca. Instead, we interpret their variations qualitatively to support the evidence from foraminiferal $\delta^{13}$C (Fig. 3) and other export productivity proxy-data (the concentration of phytoplankton-induced sterols) obtained from the same core and published in Hoff *et al.*[16] (Fig. 4) (see Discussion).

Collectively, the $\delta^{13}$C and Cd/Ca records indicate an increase in the nutrient content during the Heinrich stadials studied herein. There is a $\sim 0.5‰$ decrease in $\delta^{13}$C during the LGM and the Eemian compared with the Holocene (Fig. 3h), while the Cd/Ca values remain almost the same (Fig. 3g). The $\sim 0.5‰$ lower $\delta^{13}$C values during the LGM with almost no concomitant change in Cd/Ca may be due to the transfer of isotopically light terrestrial carbon[31], and elevated $[CO_3^{2-}]$ at the higher pH characteristic for the LGM (Fig. 3b). Elevated pH (and/or $[CO_3^{2-}]$) has been observed to lower the $\delta^{13}$C recorded by planktic foraminifera relative to seawater $\delta^{13}C_{DIC}$, but the sensitivity is species-specific and *N. pachyderma* has not yet been examined in this regard[27]. Compared with the Holocene, the lower $\delta^{13}$C values are likely due to a smaller air–sea gas exchange in response to the higher temperatures during the Eemian relative to the Holocene[33] (0.1‰ decrease in $\delta^{13}$C per 1 °C increase; ref. 34) (see Discussion below).

**Discussion**
The most striking observation from these data is the large increase in $\Delta pCO_{2cal\text{-}air}$ by $+80$ to $+100\,\mu atm$ during the final stages of HS1, HS4 and HS11. In the modern Norwegian Sea, the average $pCO_2$ at the calcification depth of *N. pachyderma* is $\sim 40\,\mu atm$ lower than in the surface ocean, where the $CO_2$ exchange with the atmosphere actually occurs (Fig. 2b). If the paleo-$pCO_2$ gradient between the calcification depth of *N. pachyderma* and the surface ocean was similar to the modern ocean ($\sim 40\,\mu atm$), the re-calculated $\Delta pCO_{2sea\text{-}air}$ values of $+40$ to $+60\,\mu atm$ during HS1, HS4 and HS11 (Fig. 3f) suggest that the Norwegian Sea, and perhaps the Nordic Seas in general, acted as a $CO_2$ source during these intervals. This is very different from the modern ocean, where the core site region is characterized by intense $CO_2$ uptake from the atmosphere (Figs 1 and 2b).

In contrast, the negative $\Delta pCO_{2sea\text{-}air}$ ($= \sim -35\,\mu atm$) during the LGM and BA interstadial could be interpreted as enhanced $CO_2$ uptake, similar to the Holocene (Fig. 3f). However, the lower

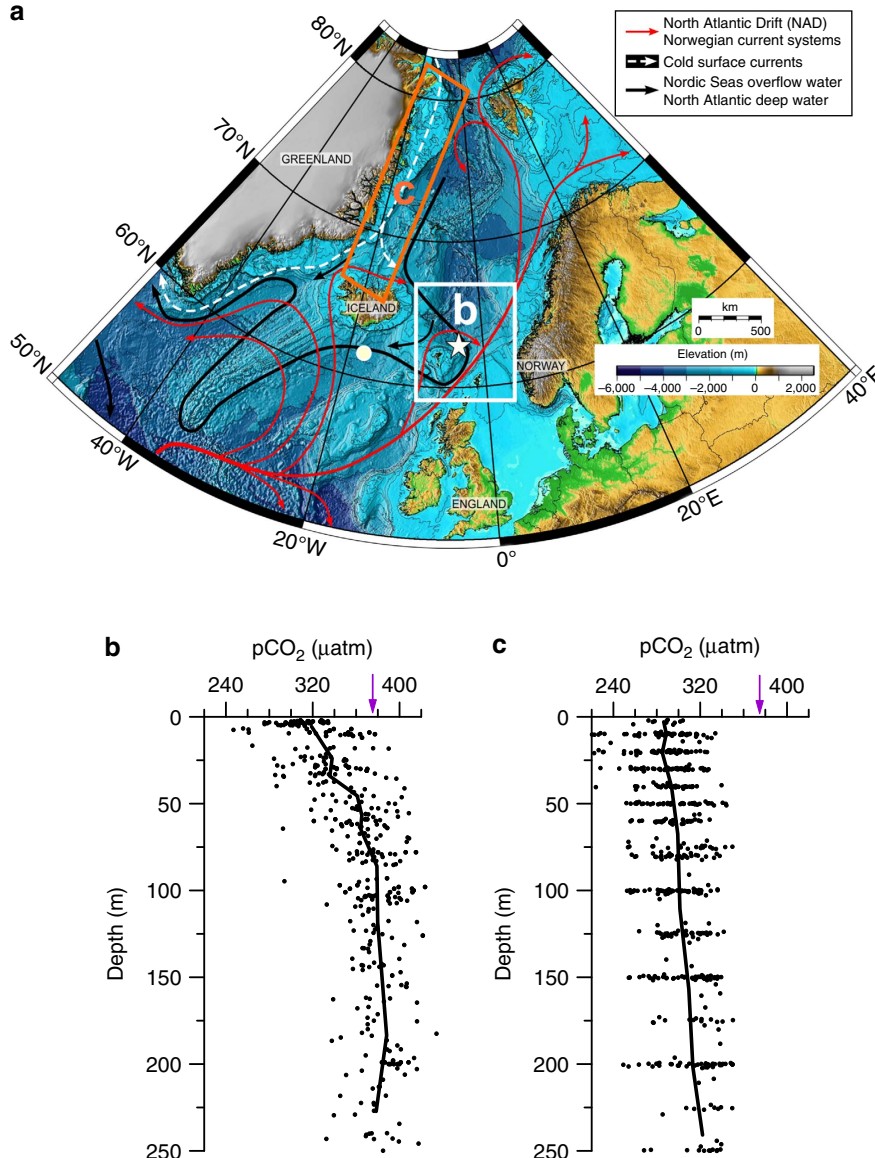

**Figure 2 | Physical oceanography and carbonate chemistry in the modern high-latitude North Atlantic.** (**a**) Map showing the major surface and bottom water currents in the northern North Atlantic and the Nordic Seas[13]. Figure modified after Ezat et al.[14]. The white star and circle indicate the location of sediment core JM11-FI-19PC (used in this study) and sediment cores studied in refs 32,37, respectively. (**b,c**) $pCO_2$-depth profiles from the Norwegian and Greenland Seas, respectively, calculated from hydrographic carbonate chemistry and nutrient data collected during 2002–2003 (ref. 21). Note that we chose only data collected during the growth seasons of N. pachyderma. The white and orange rectangles in (**a**) refer to the locations for the hydrographic sites used to construct the $pCO_2$-depth profiles in (**b,c**), respectively. The exact locations of the hydrographic sites are shown in Supplementary Fig. 4. The purple vertical arrow on the y-axes in (**b,c**) refer to the average atmospheric $pCO_2$ during 2002–2003.

aqueous $pCO_2$ values during the mid Eemian relative to the Holocene are more likely explained by a decrease in the $CO_2$ solubility because of increased sea surface temperatures. Mg/Ca temperature estimates in core JM11-FI-19PC indicate a 2 °C warming at the calcification depth of N. pachyderma[15], but faunal assemblages, which may reflect temperatures in the mixed layer, where $CO_2$ is exchanged, suggest an even greater warming up to ∼4 °C compared with the present[33].

In the discussion above, we assumed that the $pCO_2$ gradient between the calcification depth of N. pachyderma and the surface ocean (∼40 μatm) remained constant through time. We cannot provide evidence for past changes in this gradient; however, the modern spatial variability of this $pCO_2$ gradient in the Nordic Seas combined with inferred past changes in ocean circulation can provide some insights. Importantly, previous studies from the

Nordic Seas based on planktic foraminiferal assemblages[35] and sea-ice proxies ($IP_{25}$ and phytoplankton-based sterols) (ref. 16) suggest that the polar front moved towards our study area during cold stadial periods. A modern $pCO_2$-depth profile from the polar frontal zone in the Greenland Sea[21] (Fig. 2c) shows that the $pCO_2$ gradient between the surface ocean and the calcification depth of N. pachyderma ( = ∼20 μatm on average) (as well as the upper water column $pCO_2$ in general) is smaller at the polar front than in the Norwegian Sea (Fig. 2b,c). This pattern argues against the possibility that a larger than modern $pCO_2$ gradient existed between the surface ocean and the calcification depth of N. pachyderma during Heinrich stadials. Our recalculated $\Delta pCO_{2sea\text{-}air}$ (Fig. 3f) may therefore actually represent a minimum estimate of the $\Delta pCO_{2sea\text{-}air}$ during these time intervals. It is notable that earlier findings by Yu et al.[32] using evidence

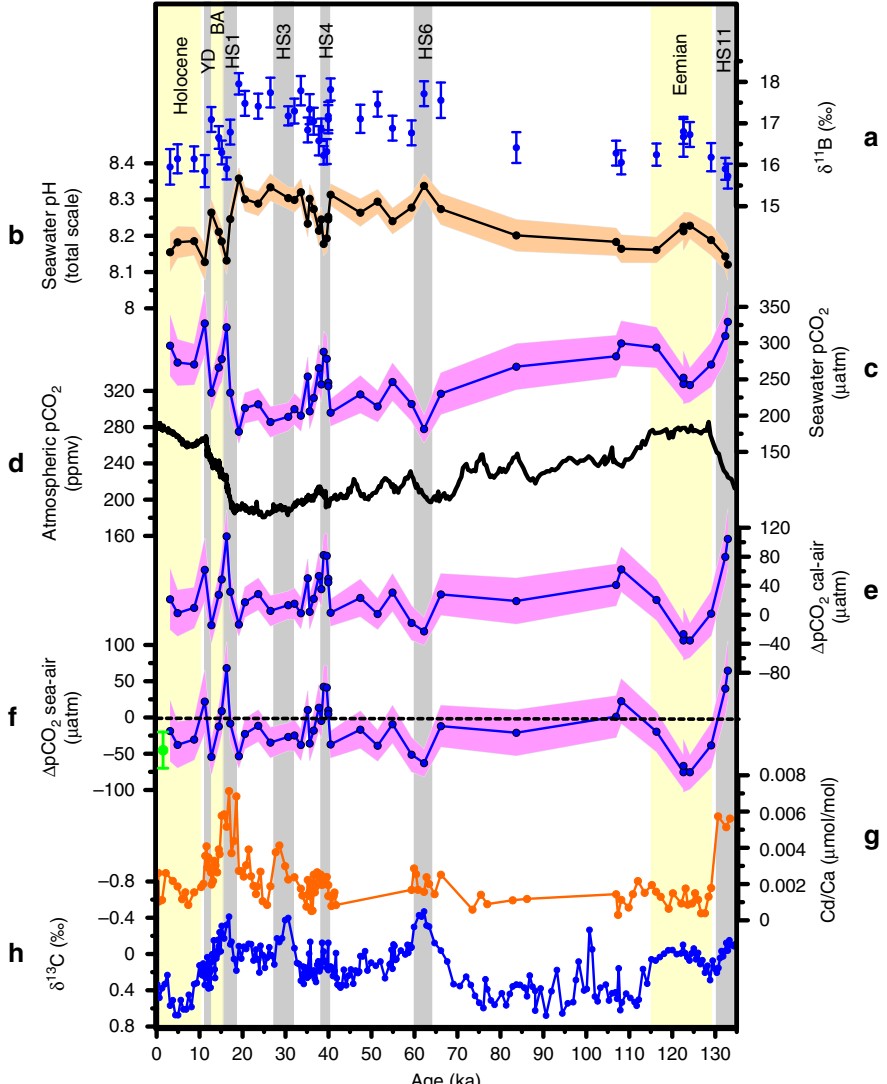

**Figure 3 | Seawater carbonate chemistry and nutrient reconstructions in sediment core JM-FI-19PC from the Norwegian Sea.** (**a**) $\delta^{11}$B measured in *N. pachyderma* with analytical uncertainty. (**b**) seawater-pH inferred from $\delta^{11}$B. (**c**) estimated seawater $pCO_2$ at the calcification depth and growth season of *N. pachyderma*. The envelope reflects the uncertainty boundaries based on the propagated error of the individual uncertainties in the parameters used to calculate $pCO_2$. (**d**) Atmospheric $pCO_2$ from Antarctic ice cores[18]. (**e**) the difference between reconstructed shallow subsurface $pCO_2$ at our site and atmospheric $pCO_2$ ($\Delta pCO_{2cal\text{-}air}$). (**f**) $\Delta pCO_{2sea\text{-}air}$ calculated as $\Delta pCO_{2cal\text{-}air}$ minus the modern $pCO_2$ gradient between the calcification depth of *N. pachyderma* (40–200 m water depth) and surface ocean (0–30 m water depth). The green circle indicates present day average $\Delta pCO_{2sea\text{-}air}$ in the Norwegian Sea[21]. (**g**) Cd/Ca measured in *N. pachyderma*. (**h**) $\delta^{13}$C measured in *N. pachyderma*.

from B/Ca and a low-resolution $\delta^{11}B_{N.\ pachyderma}$ record from the Iceland Basin, suggested that the high-latitude North Atlantic region remained a $CO_2$ sink throughout the last deglaciation. This result contrasts with our $\delta^{11}$B record despite the fact that our B/Ca record looks very similar to the B/Ca record of Yu *et al.*[32] (Supplementary Fig. 3). However, because Pleistocene planktic B/Ca records typically display large variability that rarely relates to oceanic pH variations[36], we suggest that the $\delta^{11}$B proxy is a more reliable pH proxy. The $\delta^{11}$B proxy has been validated against ice core $CO_2$ data and consistent variations in $\delta^{11}$B have been reconstructed between different core sites, where $CO_2$ is in equilibrium with the atmosphere[29,30]. Furthermore, the earlier $\delta^{11}$B study[32] does not extend beyond HS1 and may therefore fail to capture the full glacial/interglacial variability (Supplementary Fig. 3). Nevertheless, because we reconstruct air-sea disequilibrium conditions, which may be spatially variable, the discrepancy between these two $\delta^{11}$B records across HS1 (Supplementary Fig. 3) warrants additional research to further

explore the spatial extent of the high-latitude North Atlantic $pCO_2$ source during Heinrich Stadials.

The increase in $\Delta pCO_{2sea\text{-}air}$ during HS1, HS4 and HS11 in the Norwegian Sea could be the result of the following scenarios: (1) mixing with or surfacing of older water masses with accumulated $CO_2$, (2) changes in primary productivity and nutrient concentrations, (3) increased rate of sea ice formation, (4) enriched $CO_2$ content of the inflowing Atlantic water (that is, changes in the $pCO_2$ of the source water at lower latitudes) and/or (5) slowdown of deep-water formation.

Concerning scenario (1), shallow subsurface radiocarbon reconstructions from the high-latitude North Atlantic[37–39] display a prominent decrease in reservoir ages (that is, better ventilated 'young' water) at 16.5 ka, when our record shows an increase in $pCO_2$. This comparison eliminates mixing with an aged, $CO_2$-rich water mass as an explanation for our $\Delta pCO_{2sea\text{-}air}$ record. For scenario (2), the increased $\Delta pCO_{2sea\text{-}air}$ during HS1, HS4 and HS11 coincides with low $\delta^{13}$C and high Cd/Ca values,

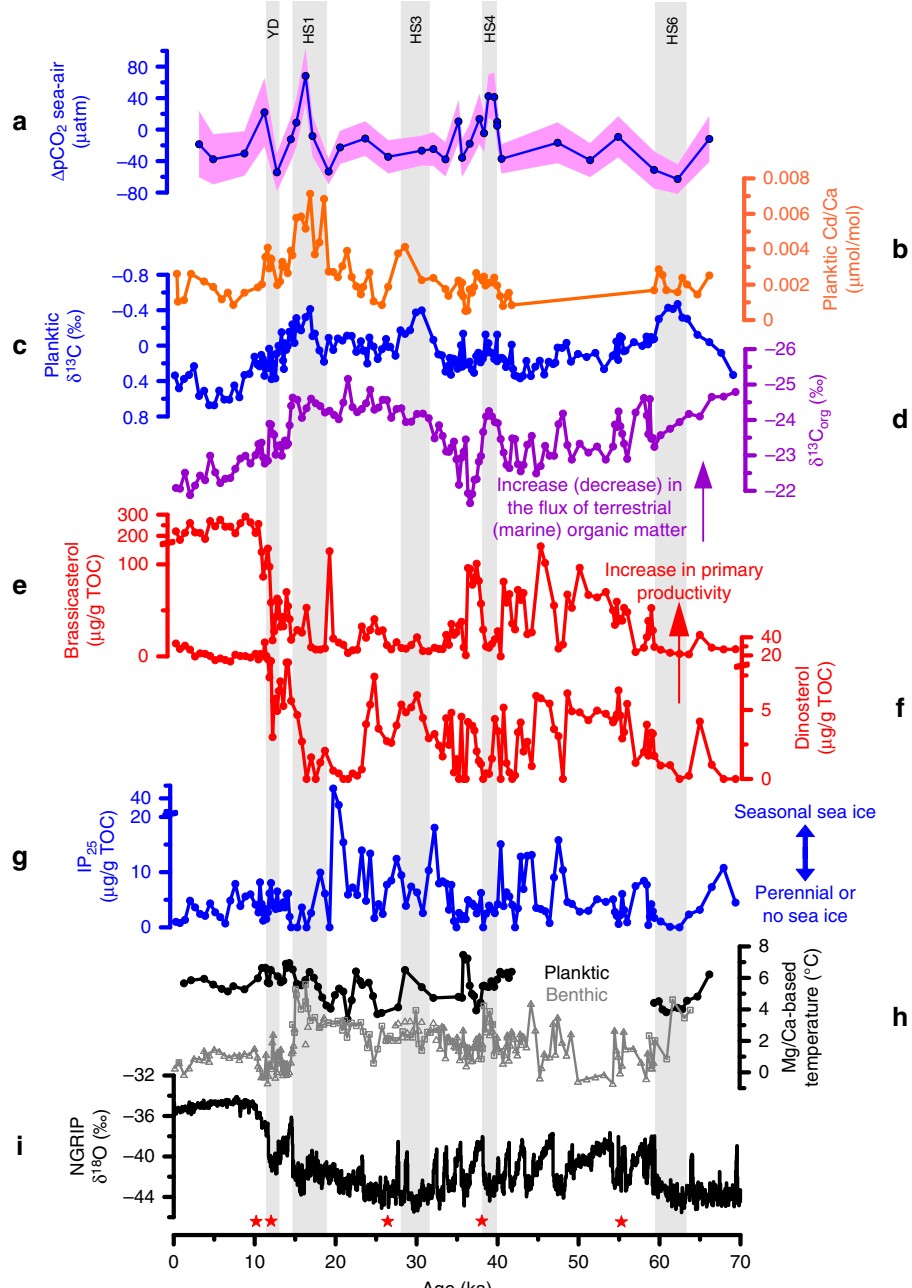

**Figure 4 | Proxy records of sediment core JM-FI-19PC plotted with North Greenland Ice Core Project δ18O values.** (**a**) ΔpCO2sea-air. (**b**) Cd/Ca measured in *N. pachyderma*. (**c**) δ13C measured in *N. pachyderma*. (**d**) δ13C measured in organic matter (δ13Corg) (ref. 16). (**e**) concentration of brassicasterol[16]. (**f**) concentration of dinosterol[16]. (**g**) C25 isoprenoid lipid (IP25) (ref. 16). High concentration of IP25 suggests presence of seasonal sea ice, whereas absence of IP25 suggests either permanent sea-ice cover (when the concentration of sterols is low) or open ocean conditions (when the concentration of sterols is high) (see Hoff *et al.*[16] for details). Note the break in the y-axes of plots **e–g**. (**h**) shallow subsurface (black) and bottom water (grey) temperature[14,15]. Bottom water temperatures are based on Mg/Ca in the benthic foraminiferal species *Melonis barleeanus* (triangles) and *Cassidulina neoteretis* (squares). Shallow subsurface temperatures are based on Mg/Ca in *N. pachyderma*. (**i**) North Greenland Ice Core Project (NGRIP) ice core δ18O values[28,67]. Red stars on the *x*-axis indicate tephra layers that are common to sediment core JM11-FI-19PC and Greenland ice cores (Supplementary Fig. 1).

so we interpret our observations as a decrease in nutrient utilization and primary production at the sea surface. A decrease in primary productivity would reduce nutrients and CO2 utilization (that is, high Cd/Ca and high pCO2), and δ13CDIC would not be elevated by preferential photosynthetic removal of 12C (that is, low foraminiferal δ13C). A decrease in the concentration of phytoplankton-induced sterols during HS4 and to some extent during HS1 (ref. 16) support the scenario of

diminished primary productivity (Fig. 4). The increase in seawater pCO2 and nutrients might also be caused by enhanced transfer of terrestrial carbon during Heinrich events and subsequent release via respiration. Hoff *et al.*[16] recorded a relative decrease in δ13Corg during HS1 and HS4 (Fig. 4d), which may reflect a combination of both decreased primary productivity (that is, decrease in the relative proportion of marine organic matter) and increased proportion of terrigenous organic matter[40].

Regarding scenario (3), studies from the modern East Greenland current region show that total dissolved inorganic carbon is rejected more efficiently than total alkalinity during sea-ice formation, causing the brines beneath the sea ice to be enriched in $CO_2$ compared with normal seawater[11]. Furthermore, modern observations from the coastal Arctic zone show substantial seasonal variations in surface ocean $pCO_2$ because of formation and melting of sea ice; with positive $\Delta pCO_{2sea-air}$ during spring and negative $\Delta pCO_{2sea-air}$ during the summer attributed to complex biogeochemical processes[41]. Because of the increased extent of sea ice during Heinrich stadials at our site[16] (Fig. 4e–g), the effect of sea ice growth/decay may have exerted a longer-term and larger-scale influence on the surface ocean $pCO_2$ in the Arctic Ocean and Nordic Seas. For scenario (4), reconstructions from the Nordic Seas of stadial ocean circulation patterns indicate a subsurface incursion of warm Atlantic water into the Nordic Seas below a well-developed halocline[14,42]. Thus, we cannot rule out that some of the $pCO_2$ increase has occurred in the source water somewhere at lower latitudes. In addition, the increase in the subsurface temperature[14,42] (Fig. 4h) may have enhanced the degradation of organic matter. Last, for scenario (5), a slow-down or cessation of deep-water formation in the Nordic Seas[14,35,42] may have promoted the $pCO_2$ increase in the shallow subsurface depth via slowing down of the carbon transfer from the sea surface to the ocean interior.

As illustrated above, several processes may have contributed to the $pCO_2$ increase during HS1, HS4 and HS11 including decreased primary productivity, increased input of terrestrial organic matter, high rate of sea ice formation and suppressed deep water formation. Conversely, during the interstadials studied herein (interstadial 8 and the BA interstadial) increased primary productivity, decreased input of terrestrial organic carbon, melting of sea ice[16] (Fig. 4) and enhanced deep water formation[14,35], resulted in the consumption and/or dilution of the $CO_2$ content. Heinrich stadials 3 and 6 are at least partially resolved in this study, but do not show similar changes in seawater carbonate chemistry as HS1, HS4 and HS11. It is notable, however, that nutrients, export productivity and sea-ice proxies suggest similar changes for all resolved Heinrich stadials (Fig. 4). We have measured $\delta^{11}B$ only for the early part of HS3 (for example, no measurements at the Cd/Ca peak), which shows a tendency towards decreasing values similar to other Heinrich stadials (Fig. 3a). During HS6, our $\delta^{11}B$ record displays an increase (that is, decrease in aqueous $pCO_2$) based on one data point (Fig. 3a). One additional difference that characterizes HS6 is the increase in $\delta^{13}C_{org}$, which suggests a relative decrease in the input of terrestrial organic matter during this event compared with other Heinrich stadials (Fig. 4). Nevertheless, higher resolution $\delta^{11}B$ records are required to assess the carbonate chemistry evolution across HS3 and HS6.

How was the oceanic $CO_2$ released to the atmosphere during HS1, HS4 and HS11 in the Norwegian Sea? The presence of thick perennial or near-perennial sea ice cover during these times[16] may have acted as a barrier for oceanic $CO_2$ outgassing. Earlier studies have suggested that a gradual build-up of a heat reservoir occurred during stadial periods because of subsurface inflow of warm Atlantic water to the Nordic Seas[14,35,42] (Fig. 4h). Surfacing of this warm water, evidenced by a large decrease in bottom water temperature[14] (Fig. 4h), occurred during the rapid transition to interstadial periods[14,42]. We therefore suggest that the $CO_2$ was released to the atmosphere, along with the advection of subsurface heat, at the terminations of the Heinrich stadials. The increases in surface $pCO_2$ in the Nordic Seas may thus have contributed to the rapid increase in atmospheric $pCO_2$ ($\sim 10$ μatm) that occurred at the terminations of some Heinrich stadials[9,43,44].

In summary, we show significant changes in the marine carbon system in the Norwegian Sea associated with well-known regional climatic anomalies during the last 135 kyr. Our data indicate that the Norwegian Sea, and possibly the broader Nordic Seas, was an area for intense $CO_2$ uptake from the atmosphere during the LGM and the interstadials investigated in this study (that is, interstadials 8 and Bølling-Allerød), similar to modern conditions, whereas it may have acted as a $CO_2$ source during the ends of HS1, HS4 and HS11. Our shallow subsurface $pCO_2$ record presents the first indication that changes in primary productivity and ocean circulation in the Nordic Seas may have played a role in the late Pleistocene variations in atmospheric $pCO_2$.

## Methods

**Age model.** The logging, scanning and sampling of the sediment core (JM-FI-19PC) are described in Ezat et al.[14] The sediment core JM-FI-19PC is aligned to the Greenland ice core NGRIP based on the identification of common tephra layers and by tuning increases in magnetic susceptibility and/or increases in benthic foraminiferal $\delta^{18}O$ values to the onset of DO interstadials in the Greenland ice cores[14–16] (Supplementary Fig. 1). In support of the reconstructed age model, eleven calibrated radiocarbon dates measured in N. pachyderma (with no attempt to correct for past changes in near-surface reservoir ages) show strong consistency with the tuned age model for the past 50 kyr (ref. 14).

**Boron isotope and minor/trace element analyses.** Only pristine N. pachyderma specimens with no visible signs of dissolution were picked from the 150 to 250 μm size fractions for boron isotope (200–450 specimens) and minor/trace element (70–160 specimens) analyses. For boron isotope measurements, the foraminifer shells were gently crushed, and cleaned following Barker et al.[45] This cleaning protocol includes clay removal, oxidative and weak acid leaching steps. Thereafter, the samples were dried and weighed to determine the amount of acid required for dissolution. Immediately before loading, samples were dissolved in ultrapure 2N HCl, and then centrifuged to separate out any insoluble mineral grains. One μl of boron-free seawater followed by an aliquot of sample solution (containing 1–1.5 ng B per aliquot) were loaded onto outgassed Rhenium filaments (zone refined), then slowly evaporated at an ion current of 0.5A and finally mounted into the mass spectrometer. Depending on sample size, five to ten replicates were loaded per sample. Boron isotopes were measured as $BO_2^-$ ions on masses 43 and 42 using a Thermo Triton thermal ionization mass spectrometer at the Lamont-Doherty Earth Observatory (LDEO) of Columbia University. Each sample aliquot was heated up slowly to $1,000 \pm 20$ °C and then 320 boron isotope ratios were acquired over $\sim 40$ min[46]. Boron isotope ratios are reported relative to the boron isotopic composition of SRM 951 boric acid standard, where $\delta^{11}B$ (‰) $= (43/42_{sample}/43/42_{standard} - 1) \times 1,000$. Analyses that fractionated $>1$‰ over the data acquisition time were discarded. The analysis of multiple replicates allows us to minimize analytical uncertainty, which is reported as 2s.e. $= 2s.d./\sqrt{n}$, where n is the number of sample aliquots analysed. The analytical uncertainty in $\delta^{11}B$ of each sample was then compared with the long-term reproducibility of an in-house vaterite standard ($\pm 0.34$‰ for $n = 3$ to $\pm 0.19$‰ for $n = 10$) and the larger of the two uncertainties is reported (Supplementary Table 1). Two samples were repeated using the oxidative-reductive cleaning procedure from Pena et al.[47] and yielded indistinguishable $\delta^{11}B$ values (Supplementary Table 1).

Trace and minor element analytical procedures followed cleaning after Martin and Lea[48] and included clay removal, reductive, oxidative, alkaline chelation (with DTPA solution) and weak acid leaching steps with slight modifications[15] from Pena et al.[47] and Lea and Boyle[49]. These modifications included rinsing samples with $NH_4OH$ (ref. 49) instead of using 0.01 N NaOH (ref. 48) as a first step to remove the DTPA solution, followed by rinsing the samples three times with cold (room temperature) MilliQ water, 5-min immersion in hot ($\sim 80$ °C) MilliQ water and two more rinses with cold MilliQ water[47]. After cleaning, the samples were dissolved in 2% $HNO_3$ and finally analysed by iCAPQ Inductively-Coupled Plasma Mass Spectrometry at LDEO. Based on repeated measurements of in-house standard solutions, the long-term precision is $<1.4$, 1.9 and 2.1% for Mg/Ca, B/Ca and Cd/Ca, respectively. Five samples were split after clay removal, reduction and oxidation steps; one half was cleaned by the full cleaning procedure, while the alkaline chelation step was omitted for the other half. This approach was applied to test the influence of the chelation step on Cd/Ca and B/Ca. The results with and without the alkaline chelation show an average difference of 0.0003 μmol mol$^{-1}$ and 5 μmol mol$^{-1}$ for Cd/Ca and B/Ca, respectively (Supplementary Table 2). The Mg/Ca values from the two cleaning methods are comparable, but two samples showed a significant decrease in Mg/Ca, Fe/Ca, Mn/Ca and Al/Ca values when the alkaline chelation step was applied (Supplementary Table 2). This might be due to a more efficient removal of contaminants that are rich in Mg, but not in Cd or B. All our Mn/Ca values from the full cleaning method are $<105$ μmol mol$^{-1}$, indicating that our results are unlikely affected by diagenetic coatings[50]. Only minor/trace element results from the full cleaning method were used in this study. All cleaning and loading steps for boron isotope and minor/trace element analyses

were done in boron-free filtered laminar flow benches and all used boron-free Milli-Q water.

**Stable isotope analyses.** Pristine specimens of the benthic foraminifera *Melonis barleeanus* ($\sim 30$ specimens, size fraction 150–315 µm) and the planktic foraminifera *N. pachyderma* ($\sim 50$ specimens, size fraction 150–250 µm) were picked for stable isotope analyses. The stable oxygen and carbon isotope analyses were performed using a Finnigan MAT 251 mass spectrometer with an automated carbonate preparation device at MARUM, University of Bremen. The external standard errors for the oxygen and carbon isotope analyses are $\pm 0.07‰$ and $\pm 0.05‰$, respectively. Values are reported relative to the Vienna Pee Dee Belemnite (VPDB), calibrated by using the National Bureau of Standards (NBS) 18, 19 and 20. The oxygen isotope data were previously presented[14–16], while the carbon isotope results are presented here for the first time (Supplementary Data 1).

**Salinity and temperature reconstructions.** We used the calcification temperature and $\delta^{18}O_{SW}$ values from Ezat *et al.*[15] based on parallel $\delta^{18}O$ and Mg/Ca measurements in *N. pachyderma* (Supplementary Data 1). Previous studies suggested that carbonate chemistry may exert a significant secondary effect on Mg/Ca in *N. pachyderma*[20]. The possible influence of secondary factors on temperature reconstructions are discussed in detail in Ezat *et al.*[15] In brief, the main effect of the secondary factors appears to be the elevated pH and carbonate ion concentration during the LGM; a correction for this effect may lower the temperatures by 0–2 °C. However, the exact effect remains uncertain[15]. Here we used the temperature and $\delta^{18}O_{SW}$ reconstructions with no correction for non-temperature factors on Mg/Ca (see section 'Propagation of error' below).

In the absence of a direct proxy for salinity, we estimated the salinity from our reconstructed $\delta^{18}O_{SW}$. There is a quasi-linear regional relationship between salinity and $\delta^{18}O_{SW}$ in the modern ocean, as both parameters co-vary because of addition/removal of freshwater[51]. However, temporal changes in the $\delta^{18}O_{SW}$ composition of freshwater sources and/or their relative contribution to a specific region, as well as changes in ocean circulation complicate using a local modern $\delta^{18}O_{SW}$-salinity relationship to infer past changes in salinity. We therefore estimate salinity using the $\delta^{18}O_{SW}$-salinity mixing line from the Norwegian Sea[51] for the Holocene and the Eemian, when the hydrological cycle and ocean circulation were likely similar to modern. For the deglacial and last glacial periods, we use the $\delta^{18}O_{SW}$-salinity mixing line[52] based on data from the Kangerdlugssuaq Fjord, East Greenland, where the dominant source of freshwater is glacial meltwater from tidewater glaciers with $\delta^{18}O_{SW}$ values ranging from $-30$ to $-20‰$. These conditions are probably more representative of the sources of glacial meltwater during deglacial and glacial times[53]. Our salinity estimates during the deglacial and last glacial periods would have been $\sim 1.5‰$ lower if we had used the modern $\delta^{18}O_{SW}$-salinity mixing line from the Norwegian Sea. Although this salinity difference may appear large, it has little consequence for our pH and pCO$_2$ reconstructions and our conclusions (see 'Sensitivity tests' below).

**pH and pCO$_2$ estimations.** The boron isotopic composition of biogenic carbonate is sensitive to seawater-pH (ref. 17), because the relative abundance and isotopic composition of the two dominant dissolved boron species in seawater, boric acid [B(OH)$_3$] and borate [B(OH)$_4^-$] changes with pH (ref. 54), and borate is the species predominantly incorporated into marine carbonates. Culture experiments with planktic foraminifera provide empirical support for using their boron isotopic composition as a pH proxy[30,55,56], but species-specific $\delta^{11}B$ offsets are also observed, which are widely ascribed to 'vital effects'[57].

Linear regressions of $\delta^{11}B_{CaCO3}$ versus $\delta^{11}B_{borate}$ relationships allow to infer $\delta^{11}B_{borate}$ from $\delta^{11}B_{CaCO3}$ (ref. 30) as follows:

$$\delta^{11}B_{borate} = (\delta^{11}B_{CaCO_3} - c)/m \quad (1)$$

where '*c*' is the intercept and '*m*' is the slope of the regression. pH can then be estimated from foraminiferal $\delta^{11}B$-based $\delta^{11}B_{borate}$ using the following equation[17]:

$$pH = pK_B - \log(-(\delta^{11}B_{SW} - \delta^{11}B_{borate})/(\delta^{11}B_{SW} - \alpha_{(B3-B4)} \times (\delta^{11}B_{borate}$$
$$- (\alpha_{(B3-B4)} - 1) \times 1,000))) \quad (2)$$

where $pK_B$ is the equilibrium constant for the dissociation of boric acid for a given temperature and salinity[58], $\delta^{11}B_{SW}$ is the $\delta^{11}B$ of seawater (modern $\delta^{11}B_{SW} = 39.61‰$; ref. 59), and $\alpha_{(B3-B4)}$ is the fractionation factor for aqueous boron isotope exchange between boric acid and borate. Klochko *et al.*[54] determined the boron isotope fractionation factor in seawater $\alpha_{(B3-B4)} = 1.0272 \pm 0.0006$.

Because $\delta^{11}B$ in the symbiont-barren *N. pachyderma* has so far only been calibrated from core top sediments, with large uncertainties and over a very limited natural pH range[32], the pH sensitivity of this species is uncertain. However, we can use evidence from other calibrated symbiont-barren planktic foraminifera species to further constrain the pH sensitivity of this species. Martínez-Botí *et al.*[60] suggested a pH sensitivity for the symbiont-barren planktic foraminifera *G. bulloides* similar to values predicted from aqueous boron isotope fractionation (that is, slope *m* in eq. (1) $= 1.074$). We therefore used a slope value of 1.074 in equation (1). In addition, we calculated the intercept $c = 2.053‰$ in equation (1) for *N. pachyderma* by calibrating our core top foraminiferal $\delta^{11}B$ to a calculated pre-industrial pH (that is, $\delta^{11}B_{borate}$). Pre-industrial pH was estimated from

modern hydrographic carbonate data (total Dissolved Inorganic Carbon 'DIC', total alkalinity, phosphate, silicate, temperature, salinity; ref. 21) from the southern Norwegian Sea (Fig. 2a, Supplementary Fig. 4), and subtracting 50 µmol kg$^{-1}$ from DIC (ref. 61) to correct for the anthropogenic CO$_2$ effect. We used the hydrographic data collected during June 2002 and from the 22nd of September to the 13th of October 2003 (that is, within the assumed calcification season of *N. pachyderma*; refs 19,20) and at our assumed calcification depth (that is, 40–120 m). This approach allows us to determine $\delta^{11}B_{borate}$ from $\delta^{11}B_{CaCO3}$ (equation 1), which can then be used to calculate pH based on equation 2.

Although the slope determined for *G. bulloides*[60] is similar to the coretop calibration of *N. pachyderma*[32], neither calibration encompasses a wide pH range, and the uncertainty of the slopes is therefore large. In contrast, laboratory culture experiments with (symbiont-bearing) planktic foraminifera cover a much wider pH-range but display a lesser pH sensitivity (slope in equation (1) $= \sim 0.7$) than predicted from aqueous boron isotope fractionation[30,55,56]. However, this difference in slope has little consequence for our pH and pCO$_2$ reconstructions. A sensitivity test using slopes $m = 1.074$ (ref. 60) and $m = 0.7$ (refs 30,55,56) shows little difference between the two estimates (see section 'Sensitivity tests' below).

If two of the six carbonate parameters (total Dissolved Inorganic Carbon (DIC), total alkalinity, carbonate ion concentration, bicarbonate ion concentration, pH and CO$_2$), are known in addition to temperature, pressure and salinity, the other parameters can be calculated[62]. We used the modern local salinity-total alkalinity relationship (Alkalinity $= 69.127 \times$ Salinity $- 116.42$, $R^2 = 0.76$, ref. 21) to estimate total alkalinity. Because weathering processes are slow and alkalinity is relatively high in the ocean, alkalinity can be considered a quasi-conservative tracer on these time scales, and we do not consider potential past changes in the salinity-total alkalinity relationship. Nonetheless, if we use the modern alkalinity-salinity relationship from the polar region as a possible analogue for our area during the last glacial, this would decrease the error in total alkalinity (because of the uncertainty in salinity) by up to 65 µmol kg$^{-1}$ (Supplementary Fig. 5). Aqueous pCO$_2$ is then calculated using CO$_2$sys.xls (ref. 63), with the equilibrium constants $K_1$ and $K_2$ from Millero *et al.*[64], $K_{SO4}$ is from Dickson[59] and the seawater boron concentration from Lee *et al.*[65].

**Sensitivity tests of pCO$_2$ reconstructions.** Supplementary Fig. 6 shows that pH and pCO$_2$ reconstructions based on very different temperature, salinity and total alkalinity scenarios are very similar and do not significantly affect the large pCO$_2$ increases during HS1, HS4 and HS11. Because the intercept '*c*' in the $\delta^{11}B_{CaCO3}$ versus $\delta^{11}B_{borate}$ calibrations (see Methods) is dependent on our choice of calcification depth for *N. pachyderma*, and corresponding selection of depths of hydrographic data to calculate the pre-industrial pH (after removing the anthropogenic carbon effect), we alternatively calculated the pre-industrial pH and the intercept '*c*' based on hydrographic data from both 50 and 200 m water depths. This sensitivity test shows that the uncertainty in the calcification depth of *N. pachyderma* has insignificant effect on the amplitude of our down core pCO$_2$ variations (Supplementary Fig. 7).

In addition, to assess the uncertainty in our pH and pCO$_2$ estimations because of the uncertainty in the $\delta^{11}B_{CaCO3}$ versus pH sensitivity in *N. pachyderma*, we recalculated the $\delta^{11}B_{borate}$ using slope value of $m = 0.7$ instead of $m = 1.074$ in equation (1) as suggested for some symbiont-bearing planktic foraminifera species[30,55,56], and re-adjusted the intercept '*c*' accordingly ($= -4.2‰$). This test shows that the uncertainty in species-specific pH-sensitivity has no effect on our pCO$_2$ reconstructions for the Heinrich stadial events, while the main difference is an increase in the glacial/interglacial pCO$_2$ by $\sim 30$ µatm, when a slope value of $m = 0.7$ is used (Supplementary Fig. 8). This brings $\Delta pCO_{2cal-air}$ for the LGM to values of $-30$ µatm (and $\Delta pCO_{2sea-air} = -70$ µatm), strengthening our conclusion about enhanced oceanic CO$_2$ uptake in our area during the LGM.

Finally, because our $\Delta pCO_{2cal-air}$ record can be biased because of errors in the age model especially for the Heinrich stadials (times with increasing atmospheric pCO$_2$), we performed a sensitivity study, in which 500 and 1,000 years were both added and subtracted from our age model (Supplementary Fig. 9). This arbitrary sensitivity study shows that such errors in the age model do not significantly affect the large increases in $\Delta pCO_{2cal-air}$ during HS1, HS4 and HS11 (Supplementary Fig. 9).

**Error propagation in pCO$_2$ reconstructions.** The uncertainty of each pCO$_2$ value in our record (Fig. 3c) is based on the propagated error of the effect of individual uncertainties in $\delta^{11}B$, calcification depth of *N. pachyderma*, temperature, salinity and total alkalinity on the pH and pCO$_2$ calculations. The error propagation ($2\sigma$) was calculated as the square root of the sum of the squared individual uncertainties. Note that total alkalinity has no effect on the pH estimations; it only affects the pCO$_2$ calculations.

The analytical uncertainty in $\delta^{11}B$ ranges from $\pm 0.22$ to $\pm 0.43‰$, which translates to $\sim \pm 10$ to $\pm 40$ µatm in pCO$_2$. The error in pCO$_2$ due to the uncertainty in the calcification depth of *N. pachyderma* is equal to $\pm 11$ µatm on average (see previous Section and Supplementary Fig. 6). The uncertainty in salinity due to the choice of different salinity-$\delta^{18}O_{SW}$ mixing models for the last glacial period and the deglaciation is $\sim \pm 1.5‰$, which translates to $\sim \pm 4$ µatm pCO$_2$. The error in total alkalinity due to the uncertainty in salinity estimations is up to $\pm 100$ µmol kg$^{-1}$, which is equivalent to $\sim \pm 9$ µatm pCO$_2$.

For the assessment of uncertainty in our temperature estimates, one should ideally consider uncertainties associated with empirical calibrations and other non-temperature factors that affect Mg/Ca in *N. pachyderma*. Because the sensitivity of Mg/Ca in *N. pachyderma* to factors other than temperature (for example, carbonate chemistry) is not known[20], we only include an error of $\pm 0.7\,°C$, based on the calibration and analytical uncertainties of Mg/Ca (see ref. 15). This uncertainty translates to $\pm 7\,\mu atm$ $pCO_2$ on average. Ezat *et al.*[15] discussed that the correction for elevated carbonate ion concentration during the LGM on Mg/Ca may lower the LGM temperature by $0–2\,°C$; however, the exact effect is very uncertain. A decrease in LGM temperatures would decrease our reconstructed $pCO_2$ values ($\sim -10\,\mu atm$ decrease per $1\,°C$ decrease), strengthening our conclusion that our study region was an intense area for $CO_2$ uptake at that time.

**Data availability.** The data generated and analysed during the current study are available along the online version of this article at the publisher's web-site.

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

## Acknowledgements

We sincerely thank J. Ruprecht, K. Esswein, U. Hoff, J. Farmer, T. Dahl, E. Ellingsen, I. Hald, K. Monsen, L. Pena, K. Allen, M. Segl and S. Pape for valuable support in the laboratory and L. Skinner, D. Thornalley, U. Hoff, J. McManus, J. Farmer and H. Spero for helpful discussions. We also thank the three anonymous reviewers for their very constructive comments and suggestions. This research was funded by the Research Council of Norway through its Centres of Excellence funding scheme, project number 223259. M.M. Ezat has also received funding from the Arctic University of Norway and the Mohn Foundation to the Paleo-CIRCUS project.

## Author contributions

M.M.E. sampled the core, performed the boron isotope analyses, cleaned the foraminiferal samples for the minor/trace analyses and wrote the first draft of the paper. T.L.R. conceived the study and contributed substantially to all aspects. B.H. supervised the boron isotope analyses, cleaning of foraminiferal samples and all carbonate chemistry calculations. All authors interpreted the results and contributed to the final manuscript.

## Additional information

**Competing financial interests:** The authors declare no competing financial interests.

