## [Peer Review File · Nature Communications]

Reviewers' comments:

Reviewer #1 (Remarks to the Author):

Review 'Episodic release of CO₂ from the high latitude North Atlantic during the last 135 ka

Ezat et al. use planktonic foraminifera boron isotope measurements, in combination with measurements of B/Ca, Mg/Ca and stable isotopes, to reconstruct seawater pCO₂ concentrations over the last 135,000 years in the high latitudes of the North Atlantic to assess whether surface waters there may have continued to be sink for atmospheric CO₂ or whether there were times when surface waters may have become a source of CO₂.

The study and the results are very interesting and topical with potential important implications. The results are also likely to add debate about the ongoing quest to understand the glacial-interglacial carbon cycle.

At the moment the manuscript is not mature enough to be readily accepted for publication. I suggest several clarifications/improvements to improve the context of the paper. Summarized it would be helpful if the authors can: clarify which data are new in their paper (apart from the d11B, B/Ca, Cd/Ca), provide all unpublished data in an appendix accompanying the paper, compare salinity reconstructions of surface record with deep benthic record (the data are there) to assess water mass stability (Ezat et al., 2014 suggest that bottom waters were 3 degrees warmer during the glacial), embed a sound reasoning of the physical circulation changes in the area, tone down the extrapolation of CO₂ source across the Norwegian Seas during these events (unless there is new evidence for this), come up with a physical mechanism to explain the observations, better explain the very low Cd/Ca data, and rewrite the abstract to better reflect the data.

With regards to surface water salinity and temperature reconstructions: why are the authors not showing higher resolution data for this core site? Hoff et al. (in review nature communications?), of which Ezat and Rasmussen are co-authors, discuss extensive sea ice formation and low salinity surface waters during Heinrich events in their manuscript featuring the same core. I think these data are crucial for the interpretation of the pCO₂ data (lines 249 and onwards), the authors should seriously consider to rewrite the current paper to include details (and data) of Hoff et al. paper in review. Perhaps it is possible for the two papers to be published in the same issue, or the Ezat paper to be published soon after the Hoff et al. paper (if it is still being considered for publication in Nature Communications). Ezat et al. discuss various drivers for the change in status of sub surface waters sea-air PCO₂ changes during the later parts of HS, including more extensive sea-ice. If the authors link their observations with observations of increased sea-ice at their location, than that will put the data in a topical context, potentially giving the paper much stronger impact. However, with this in mind the authors also need to be critical; are sea-ice areas today a source of atmospheric CO₂, and why could this be the case during the glacial?

With regards to circulation changes it is posted that there was a continuous inflow of warm Atlantic waters into the Norwegian Sea, based on benthic foraminifera sea water reconstructions at the site (Ezat et al 2014), this warm water mass is suggest to have deepened underlying a cooler, but fresher (?) water mass. There is a large wind driven component of the present day North Atlantic Current. If it were to be submerged under a cooler and fresher water mass, what would be the driving force of carrying this water into the Norwegian Sea? Another, perhaps naive, question is how thick is North Atlantic current flowing into the Norwegian Sea, and do the authors think this remained the same? The location of the core used in this study is 1.16 km; the sill depth of the Faroe Channel is 0.9 m. Unless the warm water mass was thicker than 200 m, would it not be unlikely that warm water continued to flow into the Norwegian Sea? In Rasmussen and Thomson (2004) deep waters in the Norwegian Sea are suggested to have the same salinity as North Atlantic Deep Water on the other side of the ridge, but temperatures are much cooler. Surely this means that deep waters in the Norwegian Sea are denser? If the density of deep waters in the

Norwegian Sea was bigger than in the North Atlantic/Faroe Shetland Channel, is it really likely there was an active inflow from the Faroe Shetland Channel at that depth during the glacial?

Another concern involves transparency and the correct referencing of Figures and data that already feature in other publications/submitted elsewhere:

Figure 1: data is from Takahashi et al. (2009). The Figure however does not look anything like Takahashi's maps, and so the authors really should acknowledge what kind of smoothing was carried out.

Figure 2: seems almost identical to Figure 1A of Ezat et al. (2014), with a different distribution of legend etc. The authors should add to their caption 'after Ezat et al. (2014)'.
Figure 3: shows magnetic susceptibility and stable isotope data for benthic and planktonic foraminifera of the core featured in the manuscript: JM11-FI-19PC. The benthic stable isotope data features in Ezat et al. (2014). The published benthic stable isotope data is not archived in the usual databases (pangeae.de, ncdc.noaa.gov). The supplement accompanying Ezat et al. (2014) features AMS radiocarbon ages for the core and benthic Mg/Ca data. Most of the planktonic stable oxygen isotope data feature in Hoff et al. (in review?), yet this is not acknowledged in the current study. The planktonic d13C and d18O should be provided, together with the benthic oxygen isotope data in an appendix accompanying the manuscript.

Detailed comments:

Abstract:

I am guessing that the original objective was to look at glacial-interglacial cycling of sea-air PCO₂ at this location, as the manuscript discussed this in the abstract (line 19-21), but also introduction (line 63-65). Yet no mentioning is made of glacial-interglacial variability in the abstract. Please rewrite to highlight what the manuscript actually focusses on/ what are the important observations and what are the implications?

Crucially what is currently missing from the manuscript is a physical mechanism to explain why part of the Norwegian Sea may have been a source of atmospheric CO₂ in the past.

Introduction:

First sentence is confusing. Do the authors mean high latitude North Atlantic ocean circulation. Repetition of discussion of millennial scale changes paragraph 1 and 2.

There are several hypothesis for the seesaw pattern; change 'have been' to 'may' in line 50.

Since the paper involves discussion of sources and sinks of atmospheric CO₂ within the context of glacial-interglacial and millennial CO₂ changes, show the CO₂ and Greenland ice d18O changes in your Figure 4.

Lines 61 to 63. Please also see Watson et al. (2009, Science).

Methods:

What does a seawater pCO₂ profile look like between 0 and 500 m at regions that are sources of atmospheric CO₂ today?

Please provide all the data in a supplementary information file.

Boron isotopes and trace elements:

-Please explain the large difference in boron isotopes for the Holocene between your study and the

core top measurements of Yu et al. (2013) for the Norwegian Sea?

-Please also come up with an explanation for the very low Cd/Ca values which are one order of magnitude lower than other studies. Is this a laboratory effect or location effect?

-I am bit puzzled about the seawater temperature reconstructions. The authors use the calibration of Elderfield and Ganssen (2000), which gives them Holocene values ~ 6 degrees C, similar to observations. However if you look at Jonkers et al. (2013), Mg/Ca of ~ 1.6 mmol/mol would give you that temperature, whereas the values found here during the Holocene (0.7 mmol/mol) would give much lower calcification temperatures. Please explain the differences between the methods and why the Elderfield and Ganssen (2000) approach is better.

-Please provide more detailed information about why a Fjord mixing line is used for the glacial salinity reconstructions. Is this decision based on the assumption of extended sea ice (e.g. Hoff et al., in review NC)? Is sea-ice d_{18O} the same as that of glacial meltwater?

Does the modern salinity versus total alkalinity relationship hold for the past, especially when considering changing sea-level and increased brine formation in the area?

Results:

Age model.

It would be good to have the stable isotopes (plus CO_2 ?), and ash layers in Figure 4 too to provide age constraints and to put the seawater pCO_2 changes in context with atmospheric CO_2 changes.

What does the lack of a large benthic-planktonic oxygen isotope gradient during glacial times imply?

Seawater pH and pCO_2 .

Please put modern day values of pH, seawater pCO_2 , pCO_2 sea-air in Figure 4. Although the region today is considered a sink of CO_2 , your Holocene reconstructions actually show positive values (e.g. CO_2 source)?

Cd/Ca and carbon isotopes.

Both Cd/Ca and carbon isotopes may be influenced by seawater temperature changes?

Line 173: don't Yu et al. (2013) show lower CO_2 in the Iceland basin during the LGM?

Discussion:

Line 242 to 245: The increase in nutrients, which the authors claim is evidenced by increased Cd/Ca and decreased d_{13C} , could just be the result of deeper calcification depth of N. pachyderma. In the Nordic Seas nutrients (phosphate as well as nitrate) increase from 50 to 200 μM . In this case no inferences of increased organic matter degradation or productivity would be needed.

The amount of organic carbon carried in icebergs is generally quite low and seems an unlikely candidate to suggest enhanced respiration.

Lines 266 to 269: LGM as a source of CO_2 is based on about 2 data points and the pink envelope seems to fluctuate around 0 μatm . Therefore is a bit of a strong statement (intense CO_2 uptake from the atmosphere during the LGM) for data that doesn't vary that much around that time

period. Furthermore because of the lack of other proxies (planktonic stable isotopes/GRIP) it is difficult to link specific intervals of low sea-air pCO₂ with climate events. Marine Isotope Stage 5.5 and 4 seem to be the major periods of pCO₂ sinks?

The authors describe various processes that can explain changes in seawater pCO₂, but not the mechanism behind what makes this increase cause the water mass to be a source of CO₂ to the atmosphere?

Reviewer #2 (Remarks to the Author):

The study of Ezat et al., presents new pCO₂ and nutrient records from the Norwegian Sea over the past 135 kyrs. The main conclusion of this study is that during select Heinrich stadials this area of the ocean is a source of CO₂ to the atmosphere as opposed to modern day where it acts as a sink. The study includes a significant number of new boron isotopes and trace element data from an area where there is a currently a scarcity of data and will therefore be of great interest to the community. However, there are issues with the universal calibration that is used to determine the offset between δ¹¹B N. pachyderma and δ¹¹B borate. Here, the δ¹¹B-pH relationship from symbiotic and non-symbiotic foraminifera are combined with coral data to calculate a single calibration line. This approach signals a large shift from the conventional wisdom within the community that species specific calibrations are preferable when using geochemical paleoproxies and I don't think the supplement of this paper is a suitable place to publish this new approach. In the absence of a N. pachyderma calibration (which is the motivation for constructing the universal calibration), a more conventional approach could be used of applying a number of different slopes (e.g. a symbiotic vs non-symbiotic calibration) and this would allow for the exploration of air-sea CO₂ fluxes as currently outlined in the manuscript without the need for the universal calibration. The other major area that requires further exploration is the Heinrich events where no substantial change in δ¹¹B is evident and how this new record compares to the other published δ¹¹B record from the north Atlantic (Yu et al., 2013).

The new universal calibration collapses a number of δ¹¹B vs pH calibrations on to O. universa and uses all these data points to determine a generalized calibration. There is a lack of detail in regards to both the motivation for treating the data in this way and also the methodology for constructing the calibration. Full exploration of both these areas is beyond the scope of the supplement and will most likely need to be expanded upon in a separate manuscript. Further explanation is needed regarding how the universal calibration fits into our understanding of the biological processes controlling the pH of the foraminifera/coral microenvironment i.e. given what we know about the processes of symbiont photosynthesis, foraminifera respiration, coral pH up-regulation, it is difficult to understand why one calibration fits all. In order to outline this a significant review of the previous literature would be needed in order to understand the rationale for why the approach outlined here is better than using the most appropriate species-specific calibrations. In this case no calibration is available for N. pachyderma but I am unclear about why the approach outlined here is preferable to using the biological information regarding the life habit of the foraminifera to choose the most appropriate calibration (e.g. the calibration for non-symbiotic G. bulloides (Martinez-Boti et al., 2015)).

Regarding the construction of the universal calibration more detail is needed about the method used to collapse the records onto the O. universa line. There also needs to be additional justification for this process given that the correction for corals could be as great as 8 per mil. More exploration of the data is also needed to determine the extent to which the slope is dependent on the O. universa and G. sacculifer culture experiments conducted under high pH conditions, which are arguably unlikely to be reconstructed in the palaeorecords. Some estimate of the uncertainty on the slope of the line is also needed.

Neither the d11B change across HS3 or HS6 or the record of Yu et al., 2013 are discussed due to lack of resolution in both cases. While I agree any conclusions from these records/intervals would be tentative, some discussion is warranted given the limited records of this type available in the north Atlantic. The change in the sign of the d11B change in the record of Yu et al., 2013 suggests that the change in the North Atlantic is spatially variable. Consequently some discussion is needed about the global significance of the CO₂ source identified in this study to the Heinrich events CO₂ rise. In addition the lack of change/decrease in d11B during HS3 and HS6 isn't discuss. Arguably the data resolution is not much greater in H11, however, this interval is included in the interpretation.

Points in text:

Line 77: A brief description of the calibration method would be useful in the main text

Line 127: Results should start with a description of the d11B data prior to processing.

Line 165: What are the potential reasons for the offset?

Line 168-182: This section belongs in the discussion rather than the results.

Line 189: Is there any evidence the paleo-pCO₂ gradient between the calcification depth of *N. pachyderma* and the atmosphere remained the same as the modern? Could some of the changes in $\Delta p\text{CO}_2\text{sea-air}$ be as a result of changes in surface water column structure?

Line 194: Are there any modern analogies for this magnitude of air-sea gas exchange (particularly at high latitude) to aid understanding of the processes?

Line 212: While there are limitations with the resolution of the Yu et al., 2013 record, arguably there are enough data points in HS1 to warrant some discussion of why the changes in the records are in the opposite sense.

Line 260: If the pathway for the water is from the north Atlantic, is it possible the water mass became enriched in CO₂ before it enters the Nordic sea?

Line 321: Or removal of Mg/Ca from the test as found in Barker et al., 2003?

Line 338: Include further discussion of the calibration used and issues surrounding the Mg/Ca temperature of *N. pachyderma*.

Line 367: Rephrase this sentence

Line 388: State Klochko fractionation factor with uncertainties (remove lamda)

Line 396: Add more details of the method used to collapse all empirical $\delta^{11}\text{B}\text{CaCO}_3$ calibrations.

Line 401: Consistent with Yu et al., 2013

Supplementary material:

Line 15: Is this major elemental composition data available?

Line 30: More description of calibration of the dates including any reservoir effect corrections.

Line 41: What is the uncertainty on the salinity to alkalinity relationship?

Lines 33-46: Are the uncertainties quoted 1 or 2 sigma? Also what exact method is used to propagate the uncertainty? This needs to be better explained in order for someone else to replicate.

Line 43: The issues surrounding a carbonate ion effect on Mg/Ca of *N. pachyderma* need to be expanded upon. Is it possible to determine an accurate temperature from this species?

Line 53: Why is the depth habitat effect explored as a sensitivity test rather than propagated in the uncertainty calculations?

Line 138: Expand on the rationale for adjusting the record by 1 per mil.

Line 203: The use of % here to describe $\delta^{11}\text{B}\text{CaCO}_3$ versus $\delta^{11}\text{B}\text{borate}$ sensitivity is confusing. Describe as 1 or 0.74.

Figures: All d needs to be changed to δ on the figure axes.

Figure 2: (b) and (c) rotate vertical axis labels

Figure 3: Include data points in panel (a) and (b). Panel (c) the oxygen isotope record isn't clear. Black lines denoting tephra horizons in marine records are difficult to see.

Figure 4: A panel containing the oxygen isotope record of NGRIP (Svensson et al., 2008) and from

JM-FI-19PC would aid the interpretation of this figure

Reviewer #3 (Remarks to the Author):

This is a careful study reconstructing the near-surface ocean CO₂ system in the Nordic Seas over the past 130 kyr. Based on planktic foram boron isotope data, it is likely that the core site was a source of CO₂ to the atmosphere during some prominent cold periods (HS11, HS4 and HS1) and likely remained a sink, as today, for much of the rest of the record. While I felt that the manuscript was reaching to pin-down the "why" associated with these changes, they do their best to rule out certain factors and use auxiliary Cd/Ca, C-13, and temperature proxy data (citing the authors' prior work) to discuss the most plausible scenarios. This is very novel and high quality data that deserves to be published in Nature Comms. I recommend publication after addressing some editorial comments and some substantive ones aimed at sharpening the "Discussion and Conclusions" section.

Technical point, concerning the calculation of delta-pCO₂-sea-air based on sub-surface PCO₂. The modern day delta-pCO₂-sea-air is -40 uatm (Fig. 1), yet the value based on modern *N. pachyderma* is 0. Why should we expect the profile shape of seawater PCO₂ to remain the same with time? This point is indirectly brought up in lines 188-192. "If the paleo-pCO₂ gradient between calcification depth of *N. pachyderma* and the surface ocean was similar to the modern ocean..." but it is unclear to me the justification of why one should choose this assumption over any number of possible paleo-pCO₂ depth gradients. I realize, they may be making the simplest assumption, but I think the authors should more explicitly recognize the other options (i.e., the depth gradient was larger or smaller, or in a different direction, in the past than it is today) and anticipate the impact on their results. In other words, what is the estimate and uncertainty for the actual sea surface-to-air difference across their record and not just the calcification depth-to-air difference?

Figure 1. Consider re-plotting this, zooming in on the region of interest (the size of the white star currently covers a very large area). Related idea for alternate, maybe supplemental, figure: would it be possible to look at the modern day spatial structure of the pCO₂ difference between the atmosphere and the calcification depth (~50-200 m average) using the Takahashi database? One gets a sense of this from Figure 2, but perhaps this might be an insightful figure as to the spatial heterogeneity of air-sub-sea-surface differences in the region of interest.

Figure 2. Please make it more clear which depth profile refers to which location on the map. All of this information is in the caption, but it could be made obvious in the figure alone (e.g., place "b" and "c" in the correct box, or make lines connecting the boxes to the correct sub-plots)

Line 203. "at the very sea surface". Be more precise. Would "in the mixed layer" be appropriate?

Line 234. Consider changing "is just due" to "is due solely"

Line 256. "to assess on any role of our observations for the glacial/interglacial variations..." Awkward phrasing.

Line 262. "surfacing of this warm water...was suggested to occur at the transition to the following interglacials..." The definition of "following" is unclear here. I think what is meant is "at the transition into interglacial periods".

Line 219-222. Consider numbering the hypotheses for positive PCO₂ sea-to-air flux, (1) mixing with old subsurface water, (2) changes in primary productivity and nutrient dynamics and/or (3) changes in the position of frontal systems. The proceeding discussion largely rules out (1) and (3).

The auxiliary data seems to support (2), but then the authors qualify here that they can't distinguish whether primary productivity was greater or lesser during the cold periods. That there was some change in nutrient dynamics seems to be the safer conclusion (including the cited possible terrestrial nutrient supply) than a particular change in primary productivity, so perhaps this should be emphasized as the preferred hypothesis. However, then a fourth and fifth hypothesis are presented, (4) a change in sea-ice formation and/or (5) a circulation effect of subsurface Atlantic water incursions. Present these in the initial numbered list above! And be more clear about which hypotheses are ruled out or are still plausible. I think the discussion is perfectly adequate here. It just needs a little more organization to sharpen the impact.

A- Reply to reviewer's 1 comments:

1-Ezat et al. use planktonic foraminifera boron isotope measurements, in combination with measurements of B/Ca, Mg/Ca and stable isotopes, to reconstruct seawater pCO₂ concentrations over the last 135,000 years in the high latitudes of the North Atlantic to assess whether surface waters there may have continued to be sink for atmospheric CO₂ or whether there were times when surface waters may have become a source of CO₂. The study and the results are very interesting and topical with potential important implications. The results are also likely to add debate about the ongoing quest to understand the glacial-interglacial carbon cycle. At the moment the manuscript is not mature enough to be readily accepted for publication. I suggest several clarifications/improvements to improve the context of the paper. Summarized it would be helpful if the authors can: clarify which data are new in their paper (apart from the δ¹¹B, B/Ca, Cd/Ca), provide all unpublished data in an appendix accompanying the paper,

Reply:

*We have added the requested data set of previously unpublished data (δ³C and Cd/Ca in *N. pachyderma*) in a supplementary file. The reviewer provided a more detailed comment later (comment #8) about what he/she means by 'Summarized it would be helpful if the authors can: clarify which data are new in their paper (apart from the δ¹¹B, B/Ca, Cd/Ca)'. Please see below our reply to this comment.*

2-compare salinity reconstructions of surface record with deep benthic record (the data are there) to assess water mass stability (Ezat et al., 2014 suggest that bottom waters were 3 degrees warmer during the glacial), embed a sound reasoning of the physical circulation changes in the area.

Reply:

We are now comparing the shallow subsurface (Ezat et al., 2016) and bottom water temperature (Ezat et al., 2014) reconstructions with the new data in this study in a new figure (Figure 5).

Because of the lack of constraints on the past changes of the δ⁸O_{sw} freshwater sources

(and their relative contribution) to our area, we did not attempt to reconstruct salinity changes from the $\delta^{18}\text{O}_{\text{SW}}$ reconstructions. In addition, ocean circulation changes (see Waelbroeck et al., 2011), the time transgressive nature of $\delta^{18}\text{O}_{\text{SW}}$ changes due to build up/collapse of ice sheets and glaciers (see Friedrich and Timmermann, 2012) and changes in the rate of sea ice formation (see Dokken and Jansen, 1999, Rasmussen and Thomsen, 2009) may have significant effects on the details of the salinity- $\delta^{18}\text{O}_{\text{SW}}$ relationship in the past, especially at high latitudes. The effect of these factors in shaping salinity- $\delta^{18}\text{O}_{\text{SW}}$ relationship may furthermore be different between shallow subsurface (50–200m) to intermediate water (1200 m) depths. Also, there are some intervals where we did not have enough benthic foraminifera (Ezat et al., 2014) to measure both Mg/Ca and stable isotopes from the same samples, which makes the bottom seawater $\delta^{18}\text{O}$ record relatively poor. This is also why this parameter is not shown in Ezat et al. (2014).

Given all these uncertainties, we did not attempt to calculate salinity using $\delta^{18}\text{O}_{\text{SW}}$. Despite the difficulties in reconstructing salinity, the comparison between shallow subsurface and bottom water temperatures (Fig. 5) provide insights about the water column stability.

It is important to note that any uncertainties in salinity have no significant impact on our pH and pCO₂ reconstructions (Supplementary Figure 5).

3- tone down the extrapolation of CO₂ source across the Norwegian Seas during these events (unless there is new evidence for this), come up with a physical mechanism to explain the observations, better explain the very low Cd/Ca data, and rewrite the abstract to better reflect the data.

Reply:

- *We have tried to use only suggestive language when we discussed the possibility that our area may have acted as a CO₂ source during some time periods as follows: ‘Our results suggest that the Norwegian Sea probably acted as a CO₂ source towards the ends of the Heinrich stadials HS1, HS4, and HS11..’ in the*

abstract, 'the re-calculated $\Delta pCO_{2,sea-air}$ values of 20 to 60 μatm during HS1, HS4 and HS11 suggest that the Norwegian Sea' in the discussion, and in the last paragraph 'whereas it may have acted as a CO_2 source during HS1, HS4 and HS11.'

- *Regarding the physical mechanism, very low Cd/Ca data and rewriting of the abstract, we provide more detail in comments #10, #19 and #9, respectively.*

5-With regards to surface water salinity and temperature reconstructions: why are the authors not showing higher resolution data for this core site? Hoff et al. (in review nature communications?), of which Ezat and Rasmussen are co-authors, discuss extensive sea ice formation and low salinity surface waters during Heinrich events in their manuscript featuring the same core. I think these data are crucial for the interpretation of the pCO_2 data (lines 249 and onwards), the authors should seriously consider to rewrite the current paper to include details (and data) of Hoff et al. paper in review. Perhaps it is possible for the two papers to be published in the same issue, or the Ezat paper to be published soon after the Hoff et al. paper (if it is still being considered for publication in Nature Communications). Ezat et al. discuss various drivers for the change in status of sub surface waters sea-air PCO_2 changes during the later parts of HS, including more extensive sea-ice. If the authors link their observations with observations of increased sea-ice at their location, than that will put the data in a topical context, potentially giving the paper much stronger impact. However, with this in mind the authors also need to be critical; are sea-ice areas today a source of atmospheric CO_2 , and why could this be the case during the glacial?

Reply:

We have now included all the Hoff et al. data in Figure (5) and throughout our discussion (e.g., when discussing changes in primary productivity and sea ice distribution; Lines 268–270) as these data are now published and can be referred to (Hoff et al., 2016).

are sea-ice areas today a source of atmospheric CO_2 , and why could this be the case during the glacial?

Rysgaard et al (2009) show seasonal changes in pCO_2 from ~350 μatm in winter/spring to 225 μatm in summer in the East Greenland Sea. They interpreted this observation as

indicative for sea ice growth and melting, respectively. Although $p\text{CO}_2$ increased due to sea ice formation, it was still lower than the atmospheric values (the data collected in 2006/2007, when atmospheric CO_2 was $\sim 382 \mu\text{atm}$). There is therefore no evidence for the East Greenland Sea being a source of CO_2 in the modern ocean.

In contrast, Geilfus et al. (2012) show a change in $p\text{CO}_2$ in newly formed brines in the coastal Arctic zone, from $1834 \mu\text{atm}$ in April to almost $0 \mu\text{atm}$ in June. This translates to a positive $\Delta p\text{CO}_{2\text{sea-air}}$ during spring and negative $\Delta p\text{CO}_{2\text{sea-air}}$ during the summer (Geilfus et al., 2012). The results are in agreement with many other observational studies from the coastal Arctic zone e.g., Geilfus et al., 2013 and Else et al., 2012. To stay within the allowed number of references for Nature Communications we only cite Geilfus et al., 2012.

At the end of this discussion we state that: ‘Because of the increased extent of sea ice during Heinrich stadials reaching our site (Hoff et al., 2016), the effect of sea ice growth/decay may have exerted a longer-term and larger scale influence on the surface ocean $p\text{CO}_2$ in the Arctic Ocean and Nordic Seas.’

6-With regards to circulation changes it is posted that there was a continuous inflow of warm Atlantic waters into the Norwegian Sea, based on benthic foraminifera sea water reconstructions at the site (Ezat et al 2014), this warm water mass is suggest to have deepened underlying a cooler, but fresher (?) water mass. There is a large wind driven component of the present day North Atlantic Current. If it were to be submerged under a cooler and fresher water mass, what would be the driving force of carrying this water into the Norwegian Sea? Another, perhaps naive, question is how thick is North Atlantic current flowing into the Norwegian Sea, and do the authors think this remained the same? The location of the core used in this study is 1.16 km; the sill depth of the Faroe Channel is 0.9 m. Unless the warm water mass was thicker than 200 m, would it not be unlikely that warm water continued to flow into the Norwegian Sea? In Rasmussen and Thomson (2004) deep waters in the Norwegian Sea are suggested to have the same salinity as North Atlantic Deep Water on the other side of the ridge, but temperatures are much cooler. Surely this means that deep waters in the Norwegian Sea are denser? If the density of deep waters in the

Norwegian Sea was bigger than in the North Atlantic/Faroe Shetland Channel, is it really likely there was an active inflow from the Faroe Shetland Channel at that depth during the glacial?

Reply:

In the Rasmussen and Thomsen study of 2004 it is suggested that the Fram Strait provides a modern analogue for the situation with subsurface (=intermediate Atlantic Water) flow during stadials/Heinrich events. Here Atlantic Water with a temperature ca. 5 °C continues its flow below the Polar low salinity surface water layer of the Arctic Ocean (-1.5 °C). The AW thickens to reach almost >1000–1500 m water depth and is slightly cooled in the process (2–3 °C). While it traverses in the Arctic Ocean, it loses only about 2 °C of its temperature as it reaches the Chukchi Sea as it is well-insulated from the atmosphere. A branch of the AW also returns southward in the Nordic Seas as intermediate AW (RAC) with slightly positive temperature flowing below the polar surface water of the East Greenland Current. During stadials/Heinrich events, the conditions at the southern rim of the meltwater resembled the Fram Strait today and AW became a subsurface intermediate water mass and lower latitudes than today – i.e the ‘Fram Strait’ moved south. The surface ocean temperature and salinity gradient were very steep with very warm and salty water just south of the rim of cold low salinity polar water of the meltwater (see e.g., CLIMAP, 1981). Therefore, the ‘subducted’ AW would be warmer and saltier than present-day intermediate (or subsurface) AW – and dense enough to penetrate into the Nordic Seas even if salinity here was the same as now (but it was probably slightly lower – see Rasmussen et al., 1996). Even today the overflow across the Faroe-Scotland Ridge via the Faroe-Shetland Channel is not constant, but can be periodic – in the case of a pause in outflow, the inflowing AW reaches almost to the bottom (900 m in the Channel) (S. Østerhus, pers. comm. 2003).

Impact of the wind-driven component remains speculative to our knowledge, but in the absence of paleo-wind indicators, we can only assume that wind patterns were similar to the modern patterns in the Fram Strait. The inflow to the Nordic Seas could then be both wind-driven and driven by outflow of the surface Polar water and the intermediate AW - similar to the outflow today in the Denmark Strait (but weaker because of strongly reduced convection). The principle is the most parsimonious we can think of – the only change from modern apart from the reduction in convection is the expansion of the polar

zone and polar water further south, and thus higher temperatures and salinities where the AW and the Polar water meet (cf. CLIMAP, 1981).

8-Another concern involves transparency and the correct referencing of Figures and data that already feature in other publications/submitted elsewhere:

Figure 1: data is from Takahashi et al. (2009). The Figure however does not look anything like Takahashi's maps, and so the authors really should acknowledge what kind of smoothing was carried out.

Reply:

We have not applied any smoothing. In Takahashi et al. (2009), they did not show the sea-air CO₂ difference.

Figure 2: seems almost identical to Figure 1A of Ezat et al. (2014), with a different distribution of legend etc. The authors should add to their caption 'after Ezat et al. (2014)'.
Figure 2: seems almost identical to Figure 1A of Ezat et al. (2014), with a different distribution of legend etc. The authors should add to their caption 'after Ezat et al. (2014)'.

Reply:

Done (line 798).

Figure 3: shows magnetic susceptibility and stable isotope data for benthic and planktonic foraminifera of the core featured in the manuscript: JM11-FI-19PC. The benthic stable isotope data features in Ezat et al. (2014). The published benthic stable isotope data is not archived in the usual databases (pangeae.de, ncdc.noaa.gov). The supplement accompanying Ezat et al. (2014) features AMS radiocarbon ages for the core and benthic Mg/Ca data. Most of the planktonic stable oxygen isotope data feature in Hoff et al. (in review?), yet this is not acknowledged in the current study. The planktonic d13C and d18O should be provided, together with the benthic oxygen isotope data in an appendix accompanying the manuscript.

Reply:

This is now taken care of as the two manuscripts with some of the shared data are now published (see details below). The data from Ezat et al (2014) that we use are referred to

already in the submitted manuscript. For Hoff et al, we were not sure which manuscript would be accepted first (this one or Hoff et al.) and that's why we submitted Hoff et al. as a related manuscript in the previous submission. As Hoff et al. (2016) is now published in Nature Communications, we are now referring to it for the part of the planktic $\delta^{18}\text{O}$ record (0–90 kyr time interval) that the two studies share. The same applies for Ezat et al (2016), which we submitted as a related manuscript in the previous submission. Ezat et al. (2016) is now published and we are now referring to it for the part of the stable isotope and trace element data that is common in the present study and the Ezat et al 2016-study.

In addition we are now submitting a supplemental table with all stable isotope and trace element data from Ezat et al. 2014, 2016 and this study.

Detailed comments

Abstract:

9-I am guessing that the original objective was to look at glacial-interglacial cycling of sea-air PCO₂ at this location, as the manuscript discussed this in the abstract (line 19-21), but also introduction (line 63-65). Yet no mentioning is made of glacial-interglacial variability in the abstract. Please rewrite to highlight what the manuscript actually focusses on/ what are the important observations and what are the implications?

Reply:

We have slightly revised the abstract to briefly describe the glacial/interglacial variations in atmospheric CO₂, but because of the 150-word limit we further elaborate on this in the introduction. Similarly, the main purpose, results and conclusion are briefly mentioned in the abstract.

10-Crucially what is currently missing from the manuscript is a physical mechanism to explain why part of the Norwegian Sea may have been a source of atmospheric CO₂ in the past.

Reply:

We have added a paragraph at the end of the discussion part regarding how/when this CO₂ excess may have been released to the atmosphere as follows: ‘How did the oceanic CO₂ excess release to the atmosphere during HS1, HS4 and HS11? The presence of thick perennial or near-perennial sea ice cover during these times (Hoff et al., 2016) may have acted as a barrier for oceanic CO₂ outgassing. Earlier studies have suggested that a gradual build-up of a heat reservoir occurred during stadial periods due to subsurface inflow of warm Atlantic water to the Nordic Seas (Rasmussen et al., 1996; Rasmussen and Thomsen, 2004; Ezat et al., 2014) (see Fig. 5h). Surfacing of this warm water, evidenced by large decrease in bottom water temperature (Ezat et al., 2014) (Fig. 5h), occurred during the rapid transition to interstadial periods (Rasmussen and Thomsen et al., 2004; Ezat et al., 2014). We therefore suggest that the CO₂ excess has released to the atmosphere, along with the advection of subsurface heat, at the terminations of the Heinrich stadials. The increases in surface pCO₂ in the Nordic Seas may thus have contributed to the rapid increase in atmospheric pCO₂ that occurred at the end of Heinrich stadials (Bereiter et al., 2012; Marcott et al., 2014; Bauska et al., 2016).’

Introduction:

11-First sentence is confusing. Do the authors mean high latitude North Atlantic ocean circulation. Repetition of discussion of millennial scale changes paragraph 1 and 2.

Reply:

Yes, and we have added the word ‘ocean’ (line 40). Paragraph 1 describes the millennial scale interhemispheric changes in climate, whereas paragraph 2 describes the millennial scale changes in atmospheric CO₂. We have now rephrased paragraph 2 to make it clearer.

12-There are several hypothesis for the seesaw pattern; change 'have been' to 'may' in line 50.

Reply:

Done (line 52).

13-Since the paper involves discussion of sources and sinks of atmospheric CO₂ within the context of glacial-interglacial and millennial CO₂ changes, show the CO₂ and Greenland ice δ¹⁸O changes in your Figure 4.

Reply:

Figure (4) has the atmospheric CO₂ record. We have now made a new figure (Fig. 5), in which the Greenland ice δ¹⁸O records are added.

14-Lines 61 to 63. Please also see Watson et al. (2009, Science).

Reply:

We have the added Watson et al. (2009) (line 64).

Methods:

15-What does a seawater pCO₂ profile look like between 0 and 500 m at regions that are sources of atmospheric CO₂ today?

Reply:

Overall, the high latitude North Atlantic (including the Arctic Ocean) is an intense area for CO₂ uptake from the atmosphere, but there are some areas in this region that act as a CO₂ source, at least during some seasons (see our reply to comment #5). Unfortunately, hydrographic observations from these areas are restricted to shallower than 15 m water depth (e.g., Geilfus et al., 2012; 2013, Else et al., 2012; Gross et al., 2014).

Major CO₂ source areas in the modern ocean such as the Eastern Equatorial Pacific, may not be suitable analogues for understanding the CO₂ dynamics at high latitudes in the Atlantic during the last glaciation.

16-Please provide all the data in a supplementary information file.

Reply:

We have submitted an excel file with all data used in this manuscript.

Boron isotopes and trace elements:

18-Please explain the large difference in boron isotopes for the Holocene between your study and the core top measurements of Yu et al. (2013) for the Norwegian Sea?

Reply:

The $\delta^{11}\text{B}$ values in Yu et al. (2013) were measured on multicollector inductively coupled plasma mass spectrometer, whereas we performed our $\delta^{11}\text{B}$ measurements using a thermal ionization mass spectrometer. The offsets between the two machines are well known for $\delta^{11}\text{B}$ analyses, but consistent relative variations are obtained (see Foster et al., 2013 for details). This is now explained in the figure caption of supplementary fig. (2).

19-Please also come up with an explanation for the very low Cd/Ca values which are one order of magnitude lower than other studies. Is this a laboratory effect or location effect?

Reply:

The low Cd/Ca data are indeed surprising, but the data are standardized, internally consistent, and Cd/Ca data from other locations measured in this laboratory, which were analyzed within the same time window as our samples, are consistent with independent studies from those locations. We therefore have no indication that these data are artificially lowered due to analytical issues. Furthermore, elevated Cd/Ca data can be due to contamination, but this cannot be the case for our data, as we have applied rigorous cleaning methods specifically developed for Cd/Ca, and the data are low instead of high (lines 371–376). Although previous Cd/Ca measurements from the Norwegian Sea yield higher values compared to ours (Yu et al., 2013, table 1), the glacial/interglacial trends in Cd/Ca in previous studies (Keigwin and Boyle, 1989) are similar to ours. We have expanded this discussion in the manuscript and refrain from using absolute values to infer absolute nutrient concentrations, but the trends are reasonable and consistent with variations in $\delta^{13}\text{C}$ and other export productivity proxies obtained from the same core and published in Hoff et al. (2016), which is the reason why we believe that using these data

as qualitative support (but not the basis of the argument) is reasonable.

20-I am bit puzzled about the seawater temperature reconstructions. The authors use the calibration of Elderfield and Ganssen (2000), which gives them Holocene values ~6 degrees C, similar to observations. However if you look at Jonkers et al. (2013), Mg/Ca of ~1.6 mmol/mol would give you that temperature, whereas the values found here during the Holocene (0.7 mmol/mol) would give much lower calcification temperatures. Please explain the differences between the methods and why the Elderfield and Ganssen (2000) approach is better.

Reply:

*If we use the Elderfield and Ganssen (2000) calibration as it is, Mg/Ca of 0.7 mmol/mol will yield temperature of ~3 °C, similarly to Jonkers et al (2013) (and also to Kozdon et al., 2009). However, following Thornalley et al. (2009) we have adjusted the pre-exponential constant in Elderfield and Ganssen (2000) so that the core-top samples yield temperatures that are similar to modern temperatures at the range of calcification depth of *N. pachyderma*. Given that this is now published in Ezat et al. (2016), we have summarized/rephrased this section.*

21-Please provide more detailed information about why a Fjord mixing line is used for the glacial salinity reconstructions. Is this decision based on the assumption of extended sea ice (e.g. Hoff et al., in review NC)? Is sea-ice d18O the same as that of glacial meltwater?

Reply:

The choice of this particular mixing line is based on that the freshwater sources in general (and their relative contributions) must have varied during the last glacial in our region. In general, the deglacial/glacial times were likely associated with increased influence of glacial meltwater and river runoff; e.g., Tarasov and Peltier, 2005. We have added a sentence to clarify this point (line 420).

We acknowledge the salinity reconstructions for these time intervals are very uncertain (please see our reply to comment #2); however, an uncertainty of $\pm 1.5\%$ in salinity

induce an uncertainty of only $\sim \pm 4 \mu\text{atm}$ in $p\text{CO}_2$ (line 540).

22-Does the modern salinity versus total alkalinity relationship hold for the past, especially when considering changing sea-level and increased brine formation in the area?

Reply:

Definitely no, but we lack constraints on how the salinity-total alkalinity relationship may have changed in the past at our site. We have added a line of discussion (lines 489–496) to acknowledge this uncertainty. Fortunately, an uncertainty up to $100 \mu\text{mol/kg}$ will change the $p\text{CO}_2$ by only $\sim \pm 9 \mu\text{atm}$ (line 541).

Results:

Age model.

23-It would be good to have the stable isotopes (plus CO_2 ?), and ash layers in Figure 4 too to provide age constraints and to put the seawater $p\text{CO}_2$ changes in context with atmospheric CO_2 changes.

Reply:

The atmospheric CO_2 record is already in Figure (4). We now added the Greenland ice core $\delta^{18}\text{O}$ records and the location of the ash layers to the new figure (5).

24-What does the lack of a large benthic-planktonic oxygen isotope gradient during glacial times imply?

Reply:

Many different explanations have been suggested for the glacial planktic and benthic $\delta^{18}\text{O}$ variability, which have been discussed in detail in many previous studies (e.g., Rasmussen et al., 1996; Dokken and Jansen, 1999, Hillaire-Marcel and de Vernal, 2008; Rasmussen and Thomsen, 2009; Stanford et al. 2011, Waelbroeck et al., 2011; Ezat et al., 2014). Because this study focuses on CO_2 reconstructions, and space is limited, we do not see much value in adding to the ongoing debate about the glacial planktic and

benthic $\delta^{18}O$ variability in the Nordic Seas.

For our age model, we just used the large and abrupt increase in benthic $\delta^{18}O$ (at 1035 cm core depth) that occurred before the 5e-Low/BAS-IV (at 1010 cm core depth and dated to 127 ka or 125 ka according to Rasmussen et al., 2003 and Capron et al., 2014, respectively) as a likely marker to the start of the Eemian ~130 ka.

Given that the two submitted related manuscripts (Hoff et al., 2016; Ezat et al., 2016) are now published, we have summarized this part of the age model with the appropriate references.

Seawater pH and pCO₂.

25-Please put modern day values of pH, seawater pCO₂, pCO₂ sea-air in Figure 4.

Although the region today is considered a sink of CO₂, your Holocene reconstructions actually show positive values (e.g. CO₂ source)?

Reply:

*Our Holocene $\Delta pCO_{2\text{sea-air}}$ (now defined as $\Delta pCO_{2\text{cal-air}}$; see the next paragraph) values are close to zero (average Holocene $\Delta pCO_{2\text{cal-air}}$ value = $7 \pm 38 \mu\text{atm}$). We discussed that $\Delta pCO_{2\text{sea-air}}$ (now defined as $\Delta pCO_{2\text{cal-air}}$) should not be considered as the difference between the sea surface and atmospheric pCO₂, but rather between the calcification depth of *N. pachyderma* and the atmosphere. Figure (2b) shows that the modern difference in pCO₂ between the calcification depth of *N. pachyderma* and atmosphere is almost zero ($\approx 4 \mu\text{atm}$) and thus in agreement with our Holocene $\Delta pCO_{2\text{sea-air}}$ (now defined as $\Delta pCO_{2\text{cal-air}}$).*

*To avoid any confusion regarding this point, we are now distinguishing between the pCO₂ difference between the calcification depth of *N. pachyderma* and the atmosphere ($\Delta pCO_{2\text{cal-air}}$) and pCO₂ difference between the surface ocean (mixed layer) and the atmosphere ($\Delta pCO_{2\text{sea-air}}$) (Lines 222–237).*

We have now added the modern values of the $\Delta pCO_{2\text{sea-air}}$ to Figure (4).

26-Cd/Ca and carbon isotopes.

Both Cd/Ca and carbon isotopes may be influenced by seawater temperature changes?

Reply:

The effect of temperature on $\delta^{13}\text{C}$ through its effect on air-sea gas exchange is mentioned in detail in the discussion (lines 198–201). Nevertheless, the temperature that affects foraminiferal Cd/Ca is the calcification temperature, whereas the temperature that controls the ‘air-sea gas exchange’ component in seawater $\delta^{13}\text{C}_{\text{DIC}}$ (an integrated signal in the foraminifera $\delta^{13}\text{C}$) is the temperature at the sea surface, where gas exchange with the atmosphere occurred. These two temperatures are not necessarily the same (see Lines 218–221), especially in a stratified ocean.

27-Line 173: don't Yu et al. (2013) show lower CO₂- in the Iceland basin during the LGM?

Reply:

Yu et al. (2013) show that the CO₂- was higher during the LGM (=~215 $\mu\text{mol/kg}$) in the Iceland basin and it decreased to 170–190 $\mu\text{mol/kg}$ into the Holocene (see Figure 6 and Table S1 in Yu et al., 2013). This elevated surface (near surface) ocean CO₂- during the LGM compared to the Holocene is in agreement with our results (i.e., elevated pH; Fig. 4) and previous studies from the low latitude North Atlantic (e.g., Hönisch et al., 2005, Henehan et al. 2013).

Discussion:

28-Line 242 to 245: The increase in nutrients, which the authors claim is evidenced by increased Cd/Ca and decreased $\delta^{13}\text{C}$, could just be the result of deeper calcification depth of *N. pachyderma*. In the Nordic Seas nutrients (phosphate as well as nitrate) increase from 50 to 200 m. In this case no inferences of increased organic matter degradation of productivity would be needed.

Reply:

*The calcification depth of *N. pachyderma* in the Atlantic domain in the Norwegian Sea today is deeper than in the polar and Arctic domain in the Nordic Seas (Simstich et al., 2003). Given cumulative evidence that the polar/Arctic front moved southeastward during Heinrich events (e.g., Rasmussen et al., 1996; Hoff et al., 2016), we would expect that *N. pachyderma* adopted a shallower calcification depth during Heinrich events compared to the Holocene and not the opposite and hence this strengthens our interpretations. However, we acknowledge that this is a very speculative argument and it is based on considering *N. pachyderma* studies only from the Nordic Seas. If we add studies for example from the Irminger Sea (Jonkers et al., 2010, 2013) suggesting a shallower calcification depth in the Irminger Sea than in the south/central part of the Norwegian Sea, the picture becomes even more complicated.*

Despite this controversy, we are now supporting our discussion about the changes in nutrients and primary productivity with records of phytoplankton-derived sterols from Hoff et al. (2016) (Fig. 5) as the reviewer thankfully suggested to us.

29-The amount of organic carbon carried in icebergs is generally quite low and seems an unlikely candidate to suggest enhanced respiration.

Reply:

We have rephrased the sentence and now refer to the relative decreases in $\delta^{13}C_{org}$ during some Heinrich stadials as discussed in Hoff et al. (2016).

30-Lines 266 to 269: LGM as a source of CO₂ is based on about 2 data points and the pink envelope seems to fluctuate around 0 uatm. Therefore is a bit of a strong statement (intense CO₂ uptake from the atmosphere during the LGM) for data that doesn't vary that much around that time period. Furthermore because of the lack of other proxies (planktonic stable isotopes/GRIP) it is difficult to link specific intervals of low sea-air pCO₂ with climate events. Marine Isotope Stage 5.5 and 4 seem to be the major periods of pCO₂ sinks?

Reply:

The NGRIP δ^8O and many other proxies are now plotted with these data in the new figure (5). Distinguishing between ($\Delta pCO_{2cat-air}$) and ($\Delta pCO_{2sea-air}$) should make our

argument clearer to which times our area has acted as a CO₂ source or sink.

It may be just a typo in the comment, but we have stated that our area was a sink not a source of CO₂ during the LGM.

31-The authors describe various processes that can explain changes in seawater pCO₂, but not the mechanism behind what makes this increase cause the water mass to be a source of CO₂ to the atmosphere?

Reply:

Please see our reply to comment #10.

B- Reply to reviewer's 2 comments:

1- The study of Ezat et al., presents new pCO₂ and nutrient records from the Norwegian Sea over the past 135 kyrs. The main conclusion of this study is that during select Heinrich stadials this area of the ocean is a source of CO₂ to the atmosphere as opposed to modern day where it acts as a sink. The study includes a significant number of new boron isotopes and trace element data from an area where there is a currently a scarcity of data and will therefore be of great interest to the community. However, there are issues with the universal calibration that is used to determine the offset between $\delta^{11}\text{B}$ N. pachyderma and $\delta^{11}\text{B}$ borate. Here, the $\delta^{11}\text{B}$ -pH relationship from symbiotic and non-symbiotic foraminifera are combined with coral data to calculate a single calibration line. This approach signals a large shift from the conventional wisdom within the community that species specific calibrations are preferable when using geochemical paleoproxies and I don't think the supplement of this paper is a suitable place to publish this new approach. In the absence of a N. pachyderma calibration (which is the motivation for constructing the universal calibration), a more conventional approach could be used of applying a number of different slopes (e.g. a symbiotic vs non-symbiotic calibration) and this would allow for the exploration of air-sea CO₂ fluxes as currently outlined in the manuscript without the need for the universal calibration.

Reply:

*The boron isotope community is indeed split about the boron isotope sensitivity to pH, which is the reason why we have performed a sensitivity test of using a slope value of 0.7 (from the universal calibration, which is also similar to the slope in calibrations based on symbiotic planktic foraminiferal species (e.g., Hennehan et al., 2013)) and a slope value of 1, which is suggested for the non-symbiotic foraminifer *G. bulloides* (Martinez-Boti et al., 2015)) (Supplementary Figure 8). This test shows that using either slope will not change our conclusions (lines 464–473). The supplement is indeed not the location to publish the details of such an approach, which will instead be published in a detailed book on boron proxies (Hönisch & Eggins, *Boron proxies, seawater-pH and paleo-pCO₂*, Blackwell-Wiley, in prep.), but the detail provided in the revised manuscript is sufficient to explain the approach. As already discussed in the manuscript, we find providing estimates using both slopes is much preferable over basing the estimates exclusively on the uncertain calibration slopes for symbiont-barren foraminifera species (e.g. Yu et al. 2013, Martinez-Boti et al. 2015).*

2- The other major area that requires further exploration is the Heinrich events where no substantial change in $\delta^{11}\text{B}$ is evident and how this new record compares to the other published $\delta^{11}\text{B}$ record from the north Atlantic (Yu et al., 2013).

Reply:

Two resolved Heinrich stadials in this study (HS3 and HS6) do not show similar changes in $\delta^{11}\text{B}$ as HS1, HS4 and HS11. Our $\delta^{11}\text{B}$ measurements for HS3 are limited to the early part of this event e.g., no analyses at the Cd/Ca peak (Figure 4). For HS6, we are now tentatively discussing why this event is different from other resolved Heinrich stadials (HS1, 4 and 11) in the carbonate chemistry evolution (see lines 300–311).

For the comparison between our $\delta^{11}\text{B}$ record and the $\delta^{11}\text{B}$ record in Yu et al. (2013) across Heinrich Stadial 1, we are now mentioning that the mismatch suggests that the carbonate chemistry may have evolved differently between the two sites and more and higher resolution $\delta^{11}\text{B}$ records are therefore required to constrain the spatial variability in carbonate chemistry at the high-latitude North Atlantic in the past (lines 237–252).

Consequently, we have replaced ‘the high latitude North Atlantic’ with ‘the Nordic Seas’ throughout the text, when mentioning the implications of our results e.g. ‘ $\Delta pCO_{2\text{sea-air}}$ values of 20 to 60 μatm during HS1, HS4 and HS11 suggest that the Norwegian Sea, and perhaps the Nordic Seas more generally, acted as a CO_2 source during these intervals.’

3- The new universal calibration collapses a number of d11B vs pH calibrations on to O. universa and uses all these data points to determine a generalized calibration. There is a lack of detail in regards to both the motivation for treating the data in this way and also the methodology for constructing the calibration. Full exploration of both these areas is beyond the scope of the supplement and will most likely need to be expanded upon in a separate manuscript. Further explanation is needed regarding how the universal calibration fits into our understanding of the biological processes controlling the pH of the foraminifera/coral microenvironment i.e. given what we know about the processes of symbiont photosynthesis, foraminifera respiration, coral pH up-regulation, it is difficult to understand why one calibration fits all. In order to outline this a significant review of the previous literature would be needed in order to understand the rationale for why the approach outlined here is better than using the most appropriate species-specific calibrations. In this case no calibration is available for N. pachyderma but I am unclear about why the approach outlined here is preferable to using the biological information regarding the life habit of the foraminifera to choose the most appropriate calibration (e.g. the calibration for non-symbiotic G. bulloides (Martinez-Boti et al., 2015)).

Regarding the construction of the universal calibration more detail is needed about the method used to collapse the records onto the O. universa line. There also needs to be additional justification for this process given that the correction for corals could be as great as 8 per mil. More exploration of the data is also needed to determine the extent to which the slope is dependent on the O. universa and G. sacculifer culture experiments conducted under high pH conditions, which are arguably unlikely to be reconstructed in the palaeorecords. Some estimate of the uncertainty on the slope of the line is also needed.

Reply:

We agree with the reviewer that the best choice is to use a species-specific calibration. As the reviewer hinted to, our motivation to use this approach (the combined calibration) is the lack of a well-established species-specific calibration for N. pachyderma. We also

agree that physiological processes may be important in modifying the pH environment in different species, but the evidence at hand suggests that all marine calcifiers calibrated over a wide range of pH follow the same sensitivity; those species that appear to suggest a greater sensitivity have been calibrated over an insufficient range of pH (<0.3 pH units) and with exceedingly large uncertainties of the slope that equally allow a lesser sensitivity. This problem cannot be resolved until calibrations with symbiont-barren species over a wide pH range have been accomplished, which is the reason why we provide estimates based on both sensitivities. This is the most honest approach that our current understanding of the boron isotope proxy allows, any preference for a single calibration would be less honest and entail greater uncertainty. We have now added more details about the justification of our use of the combined calibration. In addition, we are now discussing the use of this combined calibration versus the calibration from Martinez-Boti et al., 2015 (lines 464–473).

4- Neither the $\delta^{11}\text{B}$ change across HS3 or HS6 or the record of Yu et al., 2013 are discussed due to lack of resolution in both cases. While I agree any conclusions from these records/intervals would be tentative, some discussion is warranted given the limited records of this type available in the north Atlantic. The change in the sign of the $\delta^{11}\text{B}$ change in the record of Yu et al., 2013 suggests that the change in the North Atlantic is spatially variable. Consequently some discussion is needed about the global significance of the CO_2 source identified in this study to the Heinrich events CO_2 rise. In addition the lack of change/decrease in $\delta^{11}\text{B}$ during HS3 and HS6 isn't discuss. Arguably the data resolution is not much greater in H11, however, this interval is included in the interpretation.

Reply:

Please see our reply to comment #2.

Points in text:

5-Line 77: A brief description of the calibration method would be useful in the main text

We have added this as requested by the reviewer (lines 87–92).

6-Line 127: Results should start with a description of the d11B data prior to processing.

We did that (lines 140–142).

7-Line 165: What are the potential reasons for the offset?

Please see our reply to comment #19 of reviewer #1.

8-Line 168-182: This section belongs in the discussion rather than the results.

We agree, but this is sort of auxiliary discussion to the main story and it does not fit in with the flow of the discussion about the changes in the CO₂ record. When we used some of this information in the discussion, we briefly repeated it.

9-Line 189: Is there any evidence the paleo-pCO₂ gradient between the calcification depth of *N. pachyderma* and the atmosphere remained the same as the modern? Could some of the changes in $\Delta p\text{CO}_2^{\text{sea-air}}$ be as a result of changes in surface water column structure?

*We have now tried to collect some insights about how this pCO₂ gradient may have changed in the past. This is now discussed in lines (222–237) as follows: ‘In the discussion above, we assume that the pCO₂ gradient between the calcification depth of *N. pachyderma* and the surface ocean (~40 μatm) remained constant in the past. We cannot provide evidence for past changes in this pCO₂ gradient; however, the modern spatial variability of this pCO₂ gradient in the Nordic Seas combined with inferred past changes in ocean circulation might provide some insights. Importantly, previous studies from the Nordic Seas based on planktic foraminiferal assemblages (Rasmussen et al., 1996) and sea ice proxies (IP₂₅ and phytoplankton-based sterols) (Hoff et al., 2016) suggest that the polar front has moved towards our area during the cold stadial periods. A modern pCO₂-depth profile from the polar frontal zone in the Greenland Sea (Key et al., 2010) (Fig. 2c) shows that the pCO₂ gradient between the surface ocean and the calcification depth of *N. pachyderma* (as well as the upper water column pCO₂ in general) are lower at the polar front than in the Norwegian Sea (Fig. 2b, c). This pattern argues against the possibility that a larger than modern pCO₂ gradient existed between the surface ocean and the calcification depth of *N. pachyderma* during Heinrich stadials. Our recalculated*

$\Delta pCO_{2\text{sea-air}}$ may actually represent a minimum estimate of the $\Delta pCO_{2\text{sea-air}}$ at these time intervals.'

10-Line 194: Are there any modern analogies for this magnitude of air-sea gas exchange (particularly at high latitude) to aid understanding of the processes?

Please see our reply to comment #15 of reviewer #1.

11-Line 212: While there are limitations with the resolution of the Yu et al., 2013 record, arguably there are enough data points in HS1 to warrant some discussion of why the changes in the records are in the opposite sense.

We extended the discussion on this point. See our reply to comment #2.

12-Line 260: If the pathway for the water is from the north Atlantic, is it possible the water mass became enriched in CO₂ before it enters the Nordic sea?

We thank the reviewer for pointing our attention to this possibility and we have added it to the discussion (lines 256–257 and 285–290).

13-Line 321: Or removal of Mg/Ca from the test as found in Barker et al., 2003?

Both scenarios are possible, but because the decreases in Mg/Ca are associated with decreases in Fe/Ca, Mn/Ca and Al/Ca (lines 376–380), we suggested that the reason is removal of contaminants (cf, Pena et al., 2005).

14-Line 338: Include further discussion of the calibration used and issues surrounding the Mg/Ca temperature of *N. pachyderma*.

We have added more details regarding this point in 'methods' section (Lines 400–407) and in the 'error propagation' section (Lines 546–554).

15-Line 367: Rephrase this sentence

The sentence in question has been deleted.

16-Line 388: State Klochko fractionation factor with uncertainties (remove lamda)

Done (line 446).

17-Line 396: Add more details of the method used to collapse all empirical $\delta^{11}\text{BCaCO}_3$ calibrations.

Please see our replies to comments #1 and #3.

18-Line 401: Consistent with Yu et al., 2013

No, because if we use the slope value of 1 (similar to Yu et al., 2013), the 'c' value will be 0.8 (Line 516), hence it is not consistent with Yu et al., 2013.

Supplementary material:

19-Line 15: Is this major elemental composition data available?

Yes, they are published in Wastegård and Rasmussen (2014). As the age model now is published in Hoff et al. (2016) and Ezat et al. (2016), we have deleted this section.

20- Line 30: More description of calibration of the dates including any reservoir effect corrections.

Done, but this has been moved to the main text now (line 137).

21-Line 41: What is the uncertainty on the salinity to alkalinity relationship?

We have now added a line of discussion about this uncertainty (lines 489–496).

22-Lines 33-46: Are the uncertainties quoted 1 or 2 sigma? Also what exact method is used to propagate the uncertainty? This needs to be better explained in order for someone else to replicate.

We have added this information (lines 533–534).

24-Line 43: The issues surrounding a carbonate ion effect on Mg/Ca of *N. pachyderma* need to be expanded upon. Is it possible to determine an accurate temperature from this species?

We have now elaborated on this point at the 'Methods' section (Lines 400–407) and at the 'Error propagation' section (Lines 546–554).

25-Line 53: Why is the depth habitat effect explored as a sensitivity test rather than propagated in the uncertainty calculations?

We added the uncertainty in the calcification depth to the error propagation (lines 537–539).

26-Line 138: Expand on the rationale for adjusting the record by 1 per mil.

We have now omitted this adjustment and instead we used a separate y-axis for each $\delta^{11}\text{B}$ dataset (see the caption of supplementary Figure 2).

27-Line 203: The use of % here to describe $\delta^{11}\text{BCaCO}_3$ versus $\delta^{11}\text{Bborate}$ sensitivity is confusing. Describe as 1 or 0.74.

We fixed that.

28-Figures: All d needs to be changed to δ on the figure axes.

Done.

29-Figure 2: (b) and (c) rotate vertical axis labels

Done.

30-Figure 3: Include data points in panel (a) and (b). Panel (c) the oxygen isotope record isn't clear. Black lines denoting tephra horizons in marine records are difficult to see.

We fixed that.

31-Figure 4: A panel containing the oxygen isotope record of NGRIP (Svensson et al., 2008) and from JM-FI-19PC would aid the interpretation of this figure

We have added the NGRIP $\delta^{18}O$ to the new figure (5) in addition to several other proxies.

C- Reply to reviewer's 3 comments:

1-This is a careful study reconstructing the near-surface ocean CO₂ system in the Nordic Seas over the past 130 kyr. Based on planktic foram boron isotope data, it is likely that the core site was a source of CO₂ to the atmosphere during some prominent cold periods (HS11, HS4 and HS1) and likely remained a sink, as today, for much of the rest of the record. While I felt that the manuscript was reaching to pin-down the "why" associated with these changes, they do their best to rule out certain factors and use auxiliary Cd/Ca, C-13, and temperature proxy data (citing the authors' prior work) to discuss the most plausible scenarios. This is very novel and high quality data that deserves to be published in Nature Comms. I recommend publication after addressing some editorial comments and some substantive ones aimed at sharpening the "Discussion and Conclusions" section.

Technical point, concerning the calculation of delta-pCO₂-sea-air based on sub-surface PCO₂. The modern day delta-pCO₂-sea-air is -40 uatm (Fig. 1), yet the value based on modern *N. pachyderma* is 0. Why should we expect the profile shape of seawater PCO₂ to remain the same with time? This point is indirectly brought up in lines 188-192. "If the paleo-pCO₂ gradient between calcification depth of *N. pachyderma* and the surface ocean was similar to the modern ocean..." but it is unclear to me the justification of why one should choose this assumption over any number of possible paleo-pCO₂ depth gradients. I realize, they may be making the simplest assumption, but I think the authors should more explicitly recognize the other options (i.e., the depth gradient was larger or smaller, or in a different direction, in the past than it is today) and anticipate the impact on their results. In other words, what is the estimate and uncertainty for the actual sea surface-to-air difference across their record and not just the calcification depth-to-air difference?

Reply:

We agree that this an important point and it has been raised by Reviewer #2 as well.

Please see our reply to comment #9 by Reviewer #2.

2- Figure 1. Consider re-plotting this, zooming in on the region of interest (the size of the white star currently covers a very large area). Related idea for alternate, maybe supplemental, figure: would it be possible to look at the modern day spatial structure of the pCO₂ difference between the atmosphere and the calcification depth (~50-200 m average) using the Takahashi database? One gets a sense of this from Figure 2, but perhaps this might be an insightful figure as to the spatial heterogeneity of air-sub-sea-surface differences in the region of interest.

Reply:

*We have decreased the size of the star in Figure 1. Unfortunately, the Takahashi database provides predominantly sea surface pCO₂ data, and pCO₂ profiles are rather sparse. In addition, because of the uncertainty in the calcification depth of *N. pachyderma* and large pCO₂ gradients within the upper 200m (occurring at different depths in different areas in the Nordic Seas; Key et al., 2010), we prefer to show the spatial heterogeneity of the pCO₂ profile in the region of interest (Figures 2b, c). Furthermore, we are now distinguishing between the pCO₂ difference between the calcification depth of *N. pachyderma* and the atmosphere ($\Delta pCO_{2cal-air}$) and pCO₂ difference between the surface ocean (mixed layer) and the atmosphere ($\Delta pCO_{2sea-air}$) (lines), to clarify that our reconstructions represent the pCO₂ at the calcification depth of *N. pachyderma*.*

3- Figure 2. Please make it more clear which depth profile refers to which location on the map. All of this information is in the caption, but it could be made obvious in the figure alone (e.g., place "b" and "c" in the correct box, or make lines connecting the boxes to the correct sub-plots)

Reply:

We put 'b' and 'c' in the corresponding boxes in 'a' as suggested.

4- Line 203. "at the very sea surface". Be more precise. Would "in the mixed layer" be appropriate?

Yes, we have changed the sentence according to the suggestion.

5-Line 234. Consider changing "is just due" to "is due solely"

The sentence in question has been deleted.

6- Line 256. "to assess on any role of our observations for the glacial/interglacial variations..." Awkward phrasing.

The sentence in question has been deleted.

7- Line 262. "surfacing of this warm water...was suggested to occur at the transition to the following interglacials..." The definition of "following" is unclear here. I think what is meant is "at the transition into interglacial periods".

Yes and we changed it to 'at the rapid transition into the interstadial periods'.

8- Line 219-222. Consider numbering the hypotheses for positive PCO₂ sea-to-air flux, (1) mixing with old subsurface water, (2) changes in primary productivity and nutrient dynamics and/or (3) changes in the position of frontal systems. The proceeding discussion largely rules out (1) and (3). The auxiliary data seems to support (2), but then the authors qualify here that they can't distinguish whether primary productivity was greater or lesser during the cold periods. That there was some change in nutrient dynamics seems to be the safer conclusion (including the cited possible terrestrial nutrient supply) than a particular change in primary productivity, so perhaps this should be emphasized as the preferred hypothesis. However, than a fourth and fifth hypothesis are presented, (4) a change in sea-ice formation and/or (5) a circulation effect of subsurface Atlantic water incursions. Present these in the initial numbered list above! And be more clear about which hypotheses are

ruled out or are still plausible. I think the discussion is perfectly adequate here. It just needs a little more organization to sharpen the impact.

Reply:

We sincerely thank the reviewer for this suggestion to organize this part of the discussion. We have now numbered the different scenarios (Lines 253–258). We also added a more conclusive paragraph regarding the plausible scenarios to explain the recorded changes in carbonate chemistry (Lines 294–300).

References

- Bauska, T. K., et al. Carbon isotopes characterize rapid changes in atmospheric carbon dioxide during the last deglaciation. *Proc. Natl. Acad. Sci.* **113**, 3465–3470 (2016).
- Bereiter, B., et al. Mode change of millennial CO₂ variability during the last glacial cycle associated with a bipolar marine carbon seesaw. *Proc. Natl. Acad. Sci.* **109**, 9755–9760 (2012).
- Capron, E., et al. Temporal and spatial structure of multi-millennial temperature changes at high latitudes during the Last Interglacial. *Quat. Sci. Rev.* **103**, 116–133 (2014).
- Cross, J. N., et al. Annual sea-air CO₂ fluxes in the Bering Sea: Insights from new autumn and winter observations of a seasonally ice-covered continental shelf. *Journal of Geophysical Research: Oceans*, **119**, 6693–6708 (2014).
- Dokken, T. M., & Jansen, E. Rapid changes in the mechanism of ocean convection during the last glacial period. *Nature*, **401**, 458–461 (1999).
- Elderfield, H., & Ganssen, G. Past temperature and $\delta^{18}\text{O}$ of surface ocean waters inferred from foraminiferal Mg/Ca ratios. *Nature* **405**, 442–445 (2000).
- Else, B. G. T., Galley, R. J., Papakyriakou, T. N., Miller, L. A., Mucci, A., & Barber, D. (2012). Sea surface pCO₂ cycles and CO₂ fluxes at landfast sea ice edges in Amundsen Gulf, Canada. *Journal of Geophysical Research: Oceans*, 117(C9).
- Ezat, M. M., Rasmussen, T. L., & Groeneveld, J. Persistent intermediate water warming during cold stadials in the southeastern Nordic seas during the past 65 k.y. *Geology* **42**, 663–666 (2014).
- Ezat, M. M., Rasmussen, T. L., Groeneveld, J. Reconstruction of hydrographic changes in the southern Norwegian Sea during the past 135 kyr and the impact of different foraminiferal Mg/Ca cleaning protocols. *Geochem. Geophys. Geosyst.*, 17, doi:10.1002/2016GC006325 (2016).
- Friedrich, T., & Timmermann, A. Millennial-scale glacial meltwater pulses and their effect on the spatiotemporal benthic $\delta^{18}\text{O}$ variability, *Paleoceanography*, 27, doi: 10.1029/2012PA002330 (2012).
- Geilfus, N. X., et al. Dynamics of pCO₂ and related air-ice CO₂ fluxes in the Arctic coastal zone (Amundsen Gulf, Beaufort Sea). *Journal of Geophysical Research: Oceans*, 117(C9) (2012).

- Geilfus, N. X., et al. First estimates of the contribution of CaCO₃ precipitation to the release of CO₂ to the atmosphere during young sea ice growth. *Journal of Geophysical Research: Oceans*, **118**, 244-255 (2013).
- Henehan, M. J., Rae, J. W. B., Foster, G. L., et al. Calibration of the boron isotope proxy in the planktonic foraminifera *Globigerinoides ruber* for use in palaeo-CO₂ reconstruction. *Earth Planet. Sci. Lett.* **364**, 111-122 (2013).
- Hillaire-Marcel, C., & de Vernal, A. Stable isotope clue to episodic sea ice formation in the glacial North Atlantic. *Earth Planet. Sci. Lett.* **268**, 143-150 (2008).
- Hönisch, B., & Hemming, N. G. Surface ocean pH response to variations in pCO₂ through two full glacial cycles. *Earth Planet. Sci. Lett.* **236**, 305-314 (2005).
- Jonkers, L., et al. Seasonal stratification, shell flux, and oxygen isotope dynamics of left-coiling *N. pachyderma* and *T. quinqueloba* in the western subpolar North Atlantic. *Paleoceanography*, **25**, PA2204, doi:10.1002/palo.20018 (2010).
- Jonkers, L., Jiménez-Amat, P., Mortyn, P. G., & Brummer, G.-J. A. Seasonal Mg/Ca variability of *N. pachyderma* (s) and *G. bulloides*: Implications for seawater temperature reconstruction. *Earth Planet. Sci. Lett.* **376**, 137-144 (2013).
- Key, R. M., et al. The CARINA data synthesis project: introduction and overview. *Earth Syst. Sci. Data* **2**, 105-121 (2010).
- Kozdon, R., Ushikubo, T., Kita, N. T., Spicuzza, M., & Valley, J. W. Intratest oxygen isotope variability in the planktonic foraminifer *N. pachyderma*: Real vs. apparent vital effects by ion microprobe. *Chem. Geol.* **258**, 327-337 (2009).
- Marcott, S. A., et al. Centennial-scale changes in the global carbon cycle during the last deglaciation. *Nature* **514**, 616-619 (2014).
- Martínez-Botí, M. A., et al. Boron isotope evidence for oceanic carbon dioxide leakage during the last deglaciation. *Nature*, **518**, 219-222 (2015).
- McIntyre, A., & Cline, R. Seasonal reconstructions of the Earth's surface at the Last Glacial Maximum. *Geological Society of America* (1981).
- Pena, L. D., Calvo, E., Cacho, I., Eggins, S., & Pelejero, C. Identification and removal of Mn-Mg-rich contaminant phases on foraminiferal tests: Implications for Mg/Ca past temperature reconstructions. *Geochem. Geophys. Geosyst.* **6**, doi: 10.1029/2005GC000930 (2005).
- Rasmussen, T. L. & Thomsen, E. Stable isotope signals from brines in the Barents Sea: implications for brine formation during the last glaciation. *Geology*, **37**, 903-906 (2009).
- Rasmussen, T. L. & Thomsen, E. The role of the North Atlantic Drift in the millennial timescale glacial climate fluctuations. *Palaeogeogr., Palaeoclim., Palaeoecol.* **210**, 101-116 (2004).
- Rasmussen, T. L., Thomsen, E., Kuijpers, A., & Wastegård, S. Late warming and early cooling of the sea surface in the Nordic seas during MIS 5e (Eemian Interglacial). *Quat. Sci. Rev.* **22**, 809-821 (2003).
- Rasmussen, T. L., Thomsen, E., Labeyrie, L., & van Weering, T. C. E. Circulation changes in the Faeroe-Shetland Channel correlating with cold events during the last glacial period (58-10 ka). *Geology* **24**, 937-940 (1996).
- Rysgaard, S., Bendtsen, J., Pedersen, L. T., Ramløv, H., & Glud, R. N. Increased CO₂ uptake due to sea ice growth and decay in the Nordic Seas. *J. Geophys. Res.: Oceans* **114**, C09011 (2009).
- Simstich, J., Sarnthein, M., & Erlenkeuser, H. Paired δ¹⁸O signals of *Neogloboquadrina pachyderma* (s) and *Turborotalita quinqueloba* show thermal stratification structure in Nordic Seas. *Mar. Micropaleontol.* **48**, 107-125 (2003).

- Takahashi, T., et al. Climatological mean and decadal change in surface ocean pCO₂, and net sea–air CO₂ flux over the global oceans *Deep-Sea Res. Part II: Topical Studies in Oceanography* **56**, 554-577 (2009).
- Thornalley, D. J. R., H. Elderfield, and I. N. McCave (2009), Holocene oscillations in temperature and salinity of the surface subpolar North Atlantic, *Nature*, **457**, 711–714.
- Waelbroeck, C., et al. (2011), The timing of deglacial circulation changes in the Atlantic, *Paleoceanography*, *26*, doi: 10.1029/2010PA002007.
- Wastegård, S., & Rasmussen, T. L. Faroe Marine Ash Zone IV: a new MIS 3 ash zone on the Faroe Islands margin. *Geological Society, London, Special Publications*, **398**, 81-93 (2014).
- Watson, A. J., et al. Tracking the variable North Atlantic sink for atmospheric CO₂. *Science*, **326**, 1391-1393 (2009).
- Yu, J., Thornalley, D. J. R., Rae, J. W. B., & McCave, N. I. Calibration and application of B/Ca, Cd/Ca, and δ¹¹B in *Neogloboquadrina pachyderma* (sinistral) to constrain CO₂ uptake in the subpolar North Atlantic during the last deglaciation. *Paleoceanography* **28**, 237-252 (2013).

Reviewers' comments:

Reviewer #1 (Remarks to the Author):

Ezat et al. satisfactorily addressed mine, and also the other reviewers' comments (in my opinion) in their response to the reviewers' comments. The revised manuscript is much improved with the discussion part better sharpened compared with the first version.

I look forward to seeing the paper published.

Reviewer #2 (Remarks to the Author):

In the revised manuscript by Ezat et al., the majority of the suggestions have been implemented and the discussion in the manuscript now reads well. However, there are still a number of issues with the universal calibration that have not been fully addressed. While the authors have agreed the supplement is not the place to publish the details of this new calibration, the assurance that the methodology will be subsequently published in a book on boron proxies: (Hönisch & Eggins, *Boron proxies, seawater-pH and paleo-pCO₂*, Blackwell- Wiley, in prep.) is not fully satisfactory. As the book chapter is only in prep, it is likely this manuscript will be published first and therefore the original reference for the work. Given the novelty of the approach, this part of the study needs to be published separately in a manuscript where the methodology can be fully explained and the implications further explored.

Specifically, three of the points raised in the review have not been addressed:

- 1) More detail is needed about the method used to collapse the records onto the *O. universa* line.
- 2) More exploration of the data is also needed to determine the extent to which the slope is dependent on the *O. universa* and *G. sacculifer* culture experiments conducted under high pH conditions.
- 3) Further explanation is needed regarding how the universal calibration fits into our understanding of the biological processes controlling the pH of the foraminifera/coral microenvironment.

The motivation the authors give for constructing a universal calibration is that all marine calcifiers calibrated over a wide range of pH follow the same sensitivity. The (very recently) published *O. universa* calibration, however, by Hennehan et al., (2016), shows that *G. ruber* and *O. universa* have different pH sensitivities outside of uncertainty of each other. This brings into question the validity of combining all the available biogenic calibrations.

The data in this manuscript undoubtedly deserve to be published, and I would suggest that the pH change could be calculated using a number of the previously published calibrations, without the need for the universal calibration, and this would not alter the conclusions in the manuscript.

Minor comment

Lines 246-249- The phrasing here makes this section difficult to follow. Supplementary figure 2 also shows the resolution of the Yu et al., data is similar to the record in this study across HS1 so is unlikely to offer an explanation for the different trends.

Reviewer #3 (Remarks to the Author):

The authors have done a good job making improvements to the paper by responding to the reviews. In particular, the assumptions they made are more clearly stated and justified, where possible, and the discussion is better organized.

In reading the manuscript again, I suggest a few minor revisions before publication, at the discretion of the authors, to make the writing stronger and to make the presentation more impactful.

Abstract: More information about what you actually found could be included here. Roughly the first half of the abstract is quite general.

For instance, it is a good point that "surface ocean pCO₂ in key locations can therefore provide important clues...", but why is the Nordic Sea a key location? This was clearly stated in the ms (L62-64), so see if you can get this info to the abstract.

Also, the statement that "pCO₂ was more complex and dynamic than previously appreciated" probably could have been guessed before the project started. I would suggest putting instead some of your bolder conclusions such as the Nordic seas could be responsible for the increases in atm. CO₂ at the end of Heinrich events.

Nutrient proxy section: You mention how Cd/Ca is mostly related to nutrient concentrations, but there are some other controls. For planktic d¹³C, the list of potential controls is given, but the reader doesn't know which control to focus on. One could say that in this study d¹³C can help supplement Cd/Ca for nutrient concentrations, but these other controls may be important too (or whatever your preferred direction for the reader is).

Figure 3. Consider whether the age model figure is necessary, given the very similar figures presented in the recently published companion papers (Ezat et al., 2016, G3; Hoff et al., 2016 Nat. Comms). If removing this figure, consider whether you'd be able to move some of the pCO₂/pH sensitivity figures into the main text---these are quite interesting and I think impressively demonstrate the relatively tight control on carbonate proxies you have. Also, if the main reason to have figure 3 is to see the Interstadial Event labels, these could be added to the Greenland record in Fig. 5 instead.

L206. Delete the minus sign.

L259-293. Good discussion, but paragraph is too long. Consider where a good natural pause comes in.

L302. Semi-colon should change to a comma.

L308. Characterize should be characterizes.

L312. "How did the oceanic CO₂ excess release to the atmosphere". Awkward phrasing. To use "CO₂ excess", I think you should define this term earlier in the manuscript.

L321. "may thus have contributed to the rapid increase in atmospheric pCO₂ that occurred at the end of the Heinrich stadials". Give the magnitude of the increase in the atmosphere observed for these events. I think we're talking about a ~20 ppm variation? So the claim would probably be that the changes in the Nordic Seas contributed to some fraction of this 20 ppm change, i.e. 5-20ppm?

L327. "resolved interstadials 8 and B-A..." Slightly confusing phrasing. Consider using the IS8 notation here.

A- Reply to reviewer's 2 comments:

1- In the revised manuscript by Ezat et al., the majority of the suggestions have been implemented and the discussion in the manuscript now reads well. However, there are still a number of issues with the universal calibration that have not been fully addressed. While the authors have agreed the supplement is not the place to publish the details of this new calibration the assurance that the methodology will be subsequently published in a book on boron proxies: (Hönisch & Eggins, Boron proxies, seawater-pH and paleo-pCO₂, Blackwell-Wiley, in prep.) is not fully satisfactory. As the book chapter is only in prep, it is likely this manuscript will be published first and therefore the original reference for the work. Given the novelty of the approach this part of the study needs to be published separately in a manuscript where the methodology can be fully explained and the implications further explored. Specifically, three of the points raised in the review have not been addressed:

- 1) More detail is needed about the method used to collapse the records onto the *O. universa* line.
- 2) More exploration of the data is also needed to determine the extent to which the slope is dependent on the *O. universa* and *G. sacculifer* culture experiments conducted under high pH conditions.
- 3) Further explanation is needed regarding how the universal calibration fits into our understanding of the biological processes controlling the pH of the foraminifera/coral microenvironment.

The motivation the authors given for constructing a universal calibration is that all marine calcifiers calibrated over a wide range of pH follow the same sensitivity. The (very recently) published *O. universa* calibration, however, by Henehan et al., (2016), shows that *G. ruber* and *O. universa* have different pH sensitivities outside of uncertainty of each other. This brings into question the validity of combining all the available biogenic calibrations.

The data in this manuscript undoubtedly deserve to be published, and I would suggest that the pH change could be calculated using a number of the previously published calibrations, without the need for the universal calibration, and this would not alter the conclusions in the manuscript.

Reply:

We have removed the universal calibration from the manuscript as suggested by reviewer #2 and have used previously published calibration on symbiont-barren planktic foraminifera (Martínez-Botí et al., 2015). In addition, we performed a sensitivity test using the pH sensitivity that has been suggested for some symbiont-bearing planktic foraminifera (e.g., Sanyal et al., 1996, 2001; Henehan et al. 2013), which shows that the use of either sensitivity does not alter our conclusions as mentioned by reviewer 2 (please see lines 441–450 and supplementary figure 8).

2- Minor comment

Lines 246-249- The phrasing here makes this section difficult to follow. Supplementary figure

2 also shows the resolution of the Yu et al., data is similar to the record in this study across HS1 so is unlikely to offer an explanation for the different trends.

Reply:

We have slightly rephrased this part to make our arguments clearer. In particular we changed the sentence of ‘Furthermore, the earlier $\delta^{18}O$ study (Yu et al. 2013) is of lower temporal resolution than the data presented herein and may therefore fail to capture the full glacial/interglacial variability (Supplementary Fig. 3)’ to ‘Furthermore, the earlier $\delta^{18}O$ study (Yu et al., 2013) does not extend beyond HS1 and may therefore fail to capture the full glacial/interglacial variability (Supplementary Fig. 3).’ In addition, we have clarified at the end of this discussion that ‘Nevertheless, because we reconstruct air-sea disequilibrium conditions, which may be spatially variable, the discrepancy between these two $\delta^{18}O$ records across HS1 (Supplementary Fig. 3) warrants additional research to further explore the spatial extent of the high-latitude North Atlantic pCO_2 source during Heinrich Stadials.’

B- Reply to reviewer’s 3 comments:

The authors have done a good job making improvements to the paper by responding to the reviews. In particular, the assumptions they made are more clearly stated and justified, where possible, and the discussion is better organized.

In reading the manuscript again, I suggest a few minor revisions before publication, at the discretion of the authors, to make the writing stronger and to make the presentation more impactful.

1- Abstract: More information about what you actually found could be included here.

Roughly the first half of the abstract is quite general.

For instance, it is a good point that “surface ocean pCO_2 in key locations can therefore provide important clues...”, but why is the Nordic Sea a key location? This was clearly stated in the ms (L62-64), so see if you can get this info to the abstract.

Reply:

We have now made our brief mentioning of the importance of the study area in terms of modern oceanic CO_2 uptake clearer ‘Here we present a 135-kyr record of shallow

subsurface pCO₂ and nutrient levels from the Norwegian Sea, *an area of intense CO₂ uptake from the atmosphere today.*'

2- Also, the statement that "pCO₂ was more complex and dynamic than previously appreciated" probably could have been guessed before the project started. I would suggest putting instead some of your bolder conclusions such as the Nordic seas could be responsible for the increases in atm. CO₂ at the end of Heinrich events.

Reply:

We have changed the last sentence in the abstract accordingly to 'Our results suggest that the Norwegian Sea probably acted as a CO₂ source towards the ends of the Heinrich stadials HS1, HS4, and HS11 and may have contributed to the increases in atmospheric pCO₂ at these times.'

3- Nutrient proxy section: You mention how Cd/Ca is mostly related to nutrient concentrations, but there are some other controls. For planktic δ¹³C, the list of potential controls is given, but the reader doesn't know which control to focus on. One could say that in this study δ¹³C can help supplement Cd/Ca for nutrient concentrations, but these other controls may be important too (or whatever your preferred direction for the reader is).

Reply:

The relative importance of controls on foraminiferal δ¹³C varies at different time intervals. We thus chose to elaborate on the relative importance of these controls in the discussion with the support of independent evidence for past changes in seawater temperature, carbonate chemistry and nutrient content as well as for the changes in the primary productivity (e.g., lines 180-185 for the LGM, lines 185-188 for the Eemian and lines 250-262 for Heinrich stadials).

4- Figure 3. Consider whether the age model figure is necessary, given the very similar figures presented in the recently published companion papers (Ezat et al., 2016, G3; Hoff et al., 2016 Nat. Comms). If removing this figure, consider whether you'd be able to move some of the pCO₂/pH sensitivity figures into the main text---these are quite interesting and I think impressively demonstrate the relatively tight control on carbonate proxies you have. Also, if

the main reason to have figure 3 is to see the Interstadial Event labels, these could be added to the Greenland record in Fig. 5 instead.

Reply:

We have moved the age model figure to the supplements. Because we have several figures for sensitivity tests (a typical case for boron isotope studies), it may be difficult to move them to the main text. However, we have clearly described them in the text ('Methods' section).

5- L206. Delete the minus sign.

Reply:

Done.

6- L259-293. Good discussion, but paragraph is too long. Consider where a good natural pause comes in.

Reply:

We have now divided this paragraph into two paragraphs.

7- L302. Semi-colon should change to a comma.

Reply:

Done (line).

8- L308. Characterize should be characterizes.

Reply:

Done (line).

9- L312. "How did the oceanic CO₂ excess release to the atmosphere". Awkward phrasing. To use "CO₂ excess", I think you should define this term earlier in the manuscript.

Reply:

We have deleted the word ‘excess’ and changed the sentence to ‘How was the oceanic CO₂ released to the atmosphere during HS1, HS4 and HS11 in the Norwegian Sea?’.

10- L321. “may thus have contributed to the rapid increase in atmospheric pCO₂ that occurred at the end of the Heinrich stadials”. Give the magnitude of the increase in the atmosphere observed for these events. I think we’re talking about a ~20 ppm variation? So the claim would probably be that the changes in the Nordic Seas contributed to some fraction of this 20 ppm change, i.e. 5-20ppm?

Reply:

We have added the magnitude of increase the atmospheric pCO₂ at the ends of some Heinrich stadials/beginning of interstadials, which is ~10 μatm (e.g., Marcott et al. (2014) for the end of HS1/beginning of BA interstadial). We did not add quantitative information regarding the contribution of Nordic Seas to this ~10 μatm increase. It needs additional research exploring the spatial extent of our observations before we feel safe to quantify the contribution from the high-latitude areas in the North Atlantic.

11- L327. “resolved interstadials 8 and B-A...” Slightly confusing phrasing. Consider using the IS8 notation here.

Reply:

We have changed the phrasing to ‘the interstadials investigated in this study (interstadials 8 and Bolling-Allerød),...’.

References

- Ezat, M. M., Rasmussen, T. L., & Groeneveld, J. Persistent intermediate water warming during cold stadials in the southeastern Nordic seas during the past 65 k.y. *Geology* **42**, 663–666 (2014).
- Ezat, M. M., Rasmussen, T. L., Groeneveld, J. Reconstruction of hydrographic changes in the southern Norwegian Sea during the past 135 kyr and the impact of different foraminiferal Mg/Ca cleaning protocols. *Geochem. Geophys. Geosyst.*, 17, doi:10.1002/2016GC006325 (2016).

- Henehan, M. J., Rae, J. W. B., Foster, G. L., et al. Calibration of the boron isotope proxy in the planktonic foraminifera *Globigerinoides ruber* for use in palaeo-CO₂ reconstruction. *Earth Planet. Sci. Lett.* **364**, 111-122 (2013).
- Hoff, U., Rasmussen, T. L., Stein, R., Ezat, M. M., & Fahl, K. Sea ice and millennial-scale climate variability in the Nordic seas 90 ka to present. *Nat. Commun.*, 12247: doi: 10.1038/ncomms12247 (2016).
- Marcott, S. A., et al. Centennial-scale changes in the global carbon cycle during the last deglaciation. *Nature* **514**, 616–619 (2014).
- Martínez-Botí, M. A., et al. Boron isotope evidence for oceanic carbon dioxide leakage during the last deglaciation. *Nature*, **518**, 219-222 (2015).
- Sanyal, A., Bijma, J., Spero, H. J. and Lea, D. W. Empirical relationship between pH and the boron isotopic composition of *G. sacculifer*: Implications for the boron isotope paleo-pH proxy. *Paleoceanography* **16**, 515-519 (2001).
- Sanyal, A., et al. Oceanic pH control on the boron isotopic composition of foraminifera: Evidence from culture experiments. *Paleoceanography* **11**, 513-517 (1996).
- Yu, J., Thornalley, D. J. R., Rae, J. W. B., & McCave, N. I. Calibration and application of B/Ca, Cd/Ca, and $\delta^{11}\text{B}$ in *Neogloboquadrina pachyderma* (sinistral) to constrain CO₂ uptake in the subpolar North Atlantic during the last deglaciation. *Paleoceanography* **28**, 237-252 (2013).

REVIEWERS' COMMENTS:

Reviewer #2 (Remarks to the Author):

I am satisfied with the changes that have been made to the manuscript and am happy for it to be published as is.

Reviewer #2 (Remarks to the Author):

I am satisfied with the changes that have been made to the manuscript and am happy for it to be published as is.

Reply:

We are delighted to see that reviewer #2 sees that our work is now ready for publication.